# Anatomy of a High-Silica Eruption as Observed by a Local Seismic Network: The June 2011 Puyehue-Cordón Caulle Event (Southern Andes, Chile)

Daniel Basualto[1,2], Andrés Tassara[2,3,4], Jonathan Lazo Gil[5], Luis Franco-Marin[6], Carlos Cardona[6], Juan
San Martín[5], Fernando Gil-Cruz[6], Marcela Calabi-Floddy[7], Cristian Farías[8]

[1]Departamento de Ingeniería en Obras Civiles, Facultad de Ingeniería y Ciencias, Universidad de La Frontera (UFRO), Francisco Salazar #01145, Temuco, Chile.
[2]Programa de Doctorado en Ciencias Geológicas, Universidad de Concepción, Víctor Lamas #1290, Concepción, Chile.
[3]Departamento Ciencias de la Tierra, Facultad de Ciencias Químicas, Universidad de Concepción, Víctor Lamas #1290, Concepción, Chile.
[4]Núcleo Milenio CYCLO: The Seismic Cycle along Subduction Zones, Chile.
[5]Departamento de Ciencias Físicas, Universidad de La Frontera, Casilla 54-D, Temuco, Chile.
[6]Servicio Nacional de Geología y Minería (SERNAGEOMIN) - Red Nacional de Vigilancia Volcánica (RNVV) -
Observatorio Volcanológico de los Andes del Sur, (OVDAS), Rudecindo Ortega #03850, Temuco, Chile.
[7]Universidad de La Frontera, Nano-biotechnology Laboratory, Center of Plant, Soil Interaction and Natural Resources Biotechnology, Scientific and Biotechnological Bioresource Nucleus, BIOREN-UFRO, Francisco Salazar #01145, Temuco, Chile.
[8]Departamento de Geología y Obras Civiles, Facultad de Ingeniería, Universidad Católica de Temuco, Rudecindo Ortega
02950, Temuco, Chile.

*Correspondence to*: Andrés Tassara (andrestassara@udec.cl)

**Abstract**. High-silica explosive eruptions are one of the most dangerous natural phenomena, yet it is unclear which processes are involved in this infrequent kind of events. We present the first systematic characterization of near-field seismicity associated to a large high-silica eruption analyzing data recorded before, during and after the June 4[th] 2011 rhyolitic eruption of Puyehue Cordón Caulle Volcanic Complex (PCCVC). Results of a first-level data processing, developed by The Southern Andean Volcano Observatory (OVDAS) to monitor unrest and the evolution of the eruption, are complemented here with the

relocation of hypocenters into a local 1D velocity model, time-series of the *b-value* and the computation of focal mechanism. This information allows us to define several phases before and after the onset of the eruption, describing details of the space-time evolution of seismicity, defining and characterizing the seismic sources, identifying the structural-control of the magmatic intrusion and stress variations during the eruption. Our results illuminate several underlying processes, with emphasis on the possible role that basement structures had on the storage, transport and evacuation of magma. Integrating our results with

previous findings based on satellite geodesy and petrology of erupted materials, we discuss general conceptual models regarding destabilization of structurally-controlled acidic magmatic systems, the pass from unrest to eruption, changes in

eruptive style and waning phases of eruptions, with broader implications for monitoring and forecast of violent silicic eruptions.

## 1 Introduction

Volcanic eruptions involving volatile-rich, high-silica magma are one of the most dangerous natural phenomena. Fortunately, they are relatively infrequent, which means, however, that almost none of these eruptions have been witnessed by modern science. This has hampered our ability to recognize and understand instrumental near-field precursory signals of silicic eruptions that could be useful for the improvement of short-term forecasting models (Roman and Cashman, 2018; Pritchard et al., 2019). Remote satellite-based observations and far-field recorded seismicity could help to partially fill this gap, as for the

case of the 2008 rhyolitic eruption of Chaitén volcano (Wicks et al., 2011; Piña-Gauthier et al., 2013). However, direct records of instruments installed in the neighborhood of erupting silicic volcanoes are lacking.

We present the first systematic characterization of near-field seismicity occurring before, during and after a large rhyolitic eruption, the VEI 4-5 June 4th, 2011, eruption of Puyehue- Cordón Caulle Volcanic Complex (PCCVC) in the Southern Andes. These data were recorded by a seismic network installed by the Southern Andean Volcano Observatory (Observatorio

Volcanológico de los Andes del Sur, OVDAS) belonging to the Chilean Geological Survey (SERNAGEOMIN). Before 2010, this network was composed by few isolated stations located at relatively large distance (>30 km) from PCCVC (Fig. 1). The number of near-field instruments (<10 km from PCCVC) was augmented in the aftermath of the $M_W$8.8 Maule 2010 megathrust earthquake, which southern limit is located 300 km northward from PCCVC. This was motivated by the expectation of a possible post-seismic unrest of PCCVC, as could be suspected when considering that its last eruption occurred 38 hours after

the giant $M_W$9.5 Valdivia 1960 earthquake (e.g. Lara et al., 2004; Eggert and Walter, 2009).

In this study we combine relevant information published by previous authors and results of a first-level processing of the seismic data (earthquake count and classification, magnitude determination) to describe the chronology of the eruption. This is complemented by results of a second-level analysis (hypocenter relocation, focal mechanisms, *b-value* estimate) that are then used to discuss the main processes occurring during the different phases of the eruption. We compare the observed

evolution of seismicity with conceptual models of high-silica volcanic systems regarding fundamental processes like the mechanisms of magmatic destabilization, the pass from unrest to eruption, changes in eruption style and waning of magmatic flux at the end of the eruption. The evaluation of these conceptual models under the light of our results has broader implications for volcanic hazard assessment and short-term forecasting of violent rhyolitic eruptions.

## 2 Geologic setting of PCCVC and its 2011 eruption

The possible earthquake triggering of the 1960 eruption was likely facilitated by the primary structural control exerted by regional basement faults on the magmatic plumbing system at PCCVC (Lara et al., 2004; Sepúlveda et al., 2005; Singer et al., 2008). The volcanic complex is composed by three main edifices located along a NW-oriented long-live basement structure (Fig. 1): Cordillera Nevada Volcano at the NW extreme, Puyehue stratovolcano at the SE extreme, and Cordón Caulle graben connecting both extremes along a tectonic depression bounded by NW-SE fissures. Puyehue Volcano is itself located at the

intersection of this NW long-live basement structure with a major 1200 km long NNE-oriented structural system known as the Liquiñe-Ofqui Fault Zone (LOFZ). This system accommodates the dextral trench-parallel component of oblique convergence between Nazca and South American plates and it is genetically linked to the Southern Volcanic Zone (SVZ) of the Andes (Cembrano and Lara, 2009). The NW-oriented basement structure controlling the architecture of PCCVC belongs to the Andean Transverse Fault Systems (ATFS; Sanchez et al., 2013), a group of large basement structures segmenting the Southern

Andes that are inherited since the Upper Paleozoic (Piquer et al., 2021).

This structural setting means that PCCVC is dominated by transpressional stresses during the interseismic phase of the megathrust seismic cycle, which characterize the long-term crustal deformation. This explains the Late Pleistocene to Holocene evolution of the complex from a bulk basaltic composition to dacitic-rhyolitic as a consequence of prevailing closed conditions of a long-lived reservoir, where magma can differentiate forming a high-silica melt on top of a crystal-rich mush zone (Lara

et al., 2006b; Singer et al., 2008; Cembrano and Lara, 2009; Delgado, 2021). In addition to this presumed structural control on long-term evolution (Lara et al., 2006a), the last three eruptions (1921-1922, 1960 and 2011) evacuated almost the same rhyodacitic magma (67-74% $SiO_2$; Castro et al., 2013; Jay et al., 2014, Alloway et al., 2015) and were all connected to the structural borders of Cordón Caulle graben (Fig. 1). Into this framework, the analysis of the 2011 eruption from both a petrological-textural perspective (Castro et al., 2013, 2016; Alloway et al., 2015; Schipper et al., 2021) and modeling of

geodetic data (Jay et al., 2014; Wendt et al., 2016; Delgado et al., 2019 and 2021; Novoa et al., 2022) suggest that it was coeval with activity of faults serving as feeder dykes connecting a 5-8 km depth magmatic reservoir to the eruption vent.

The detailed seismicity-based characterization of the eruption presented here, allows us to independently complement these previous findings, supporting the structural control on eruptive behavior and contributing with novel information about the evolution of seismic activity during a high-silica eruption.


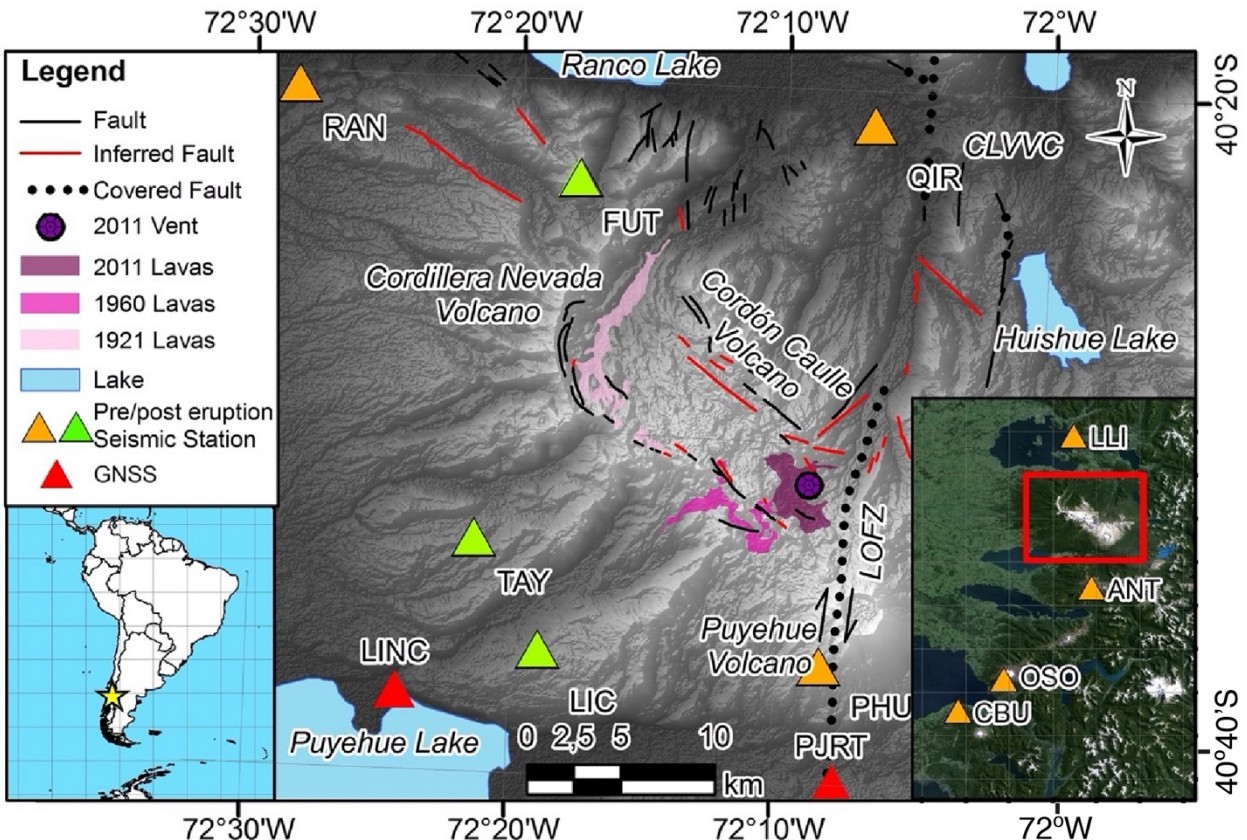

**Figure 1.** Location of Puyehue-Cordón Caulle Volcanic Complex (PCCVC) in the Southern Volcanic Zone (SVZ) of the Andes of Southern Chile. On the topographic map, we mark main geological structures, noting the Liquiñe-Ofqui Fault Zone LOFZ (Lara et al., 2006), the graben between Cordillera Nevada Volcano and Puyehue Volcano, lava flows generated by the 1921, 1960 and 2011 eruptions, the 2011 open vent and location of seismic stations (orange triangles for pre-eruption and green triangle for post-eruption location). The inset at the right bottom corner (map extracted from Google Earth) shows location of far-field seismic stations, with the red outlined square marking the location of the study area. The red triangles in the bottom show two GNSS cited in Wendt et al. (2016). CLVVC is Carran-Los Venados Volcanic Complex.

## 3 Data and methods

### 3.1 Seismological network

The seismic network considered in this study (Fig. 1) was formed mostly by Reftek151-30A broadband (0.03-50 Hz) seismic sensors connected to Reftek130B digitizers (the exception is station PHU being instrumented with a Güralp 6TD 30 second seismic sensor and a 6TD digitizer). Four of these stations (LLI, ANT, OSO, CBU) were located in the far field (35 to 80 km distance from PCCVC; Fig. 1) and form part of the regional monitoring network of OVDAS as consolidated before 2011. Other 3 stations (PHU, RAN, QIR) were installed between February and March 2011 within 10-15 km to the vent finally opened by the June 4th eruption. Up to 3 days after the eruption onset, other three stations (FUT, TAY and LIC) were installed in the southern and western flank of PCCVC. The sampling rate of seismic data was 100 samples per second. All the stations

were connected by telemetry and internet to the OVDAS observatory located in Temuco city (approx. 200 km north-westward) where a continuous flow of data was established.

## 3.2 First-level data processing and seismic signal classification

A first level of data processing was used to characterize the unrest and eruptive processes. This considers the manual
recognition of individual seismic events from the continuous seismic signal and extraction of basic information for each of them as amplitude, duration and picking of wave phases. Each recognized event was classified following the scheme of Lahr et al. (1994) and Chouet (2003), which is based on the waveform appearance on the seismic records. We show in Fig. 2 some examples of the seismic signal used to establish this classification scheme, which considers four types of volcanic seismic signals.


Volcano-Tectonic (VT, Fig. 2A) events have clear arrivals of the P and S waves, high frequencies (to 15 Hz) and are related to brittle fracturing of rocks. Long period (LP, Fig. 2B) and very long period (VLP, Fig. 2D) events usually have an emergent onset on a P body wave arrivals with a concentration of energy at low frequencies (to < 5 Hz for LP, and < 0.5 Hz for VLP) and showing long duration waveforms (>30 sec) with a single frequency coda tail. This pattern is commonly interpreted as
associated to fluids movement or pressures changes of volcanic conduits, although some authors also favor slow-rupture events in the volcanic edifice as a possible source (Bean et al., 2014). Hybrid (HB, Fig. 2C) events show a combination of features of VT and LP signals, with a predominance of impulsive compressive polarities and high frequency at the beginning of the signal that are followed by a low frequency coda, which suggest a possible fracturing of rock during the migration of magma and/or fluids on the way to surface. Tremors (TR) have an origin similar to LP events but with a sustained excitation through time
that can last by hours to months and most of them have a restricted frequency range between 0.5 to 9 Hz with a variety of emerging patterns (McNutt, 1992). Fig. 2E shows an example of the spasmodic tremor signal recorded during the explosive phase of the eruption that is characterized by a multi-frequency character (0.5-4 Hz) and a continuous oscillation of its amplitude. This is in contrast with the quasi-harmonic tremor recorded during the effusive phase of the eruption, which is exemplified in Fig. 2F. This tremor shows an energetic dominant frequency of 1 Hz with a second peak around 2 Hz and has
a modulated pattern during most of the recorded time period. As the tremors were occurring simultaneously with the occurrence of VT and HB that are characterized by short-duration high-frequency events, we isolated the tremor signal applying a Butterworth filter around the relevant low frequencies.


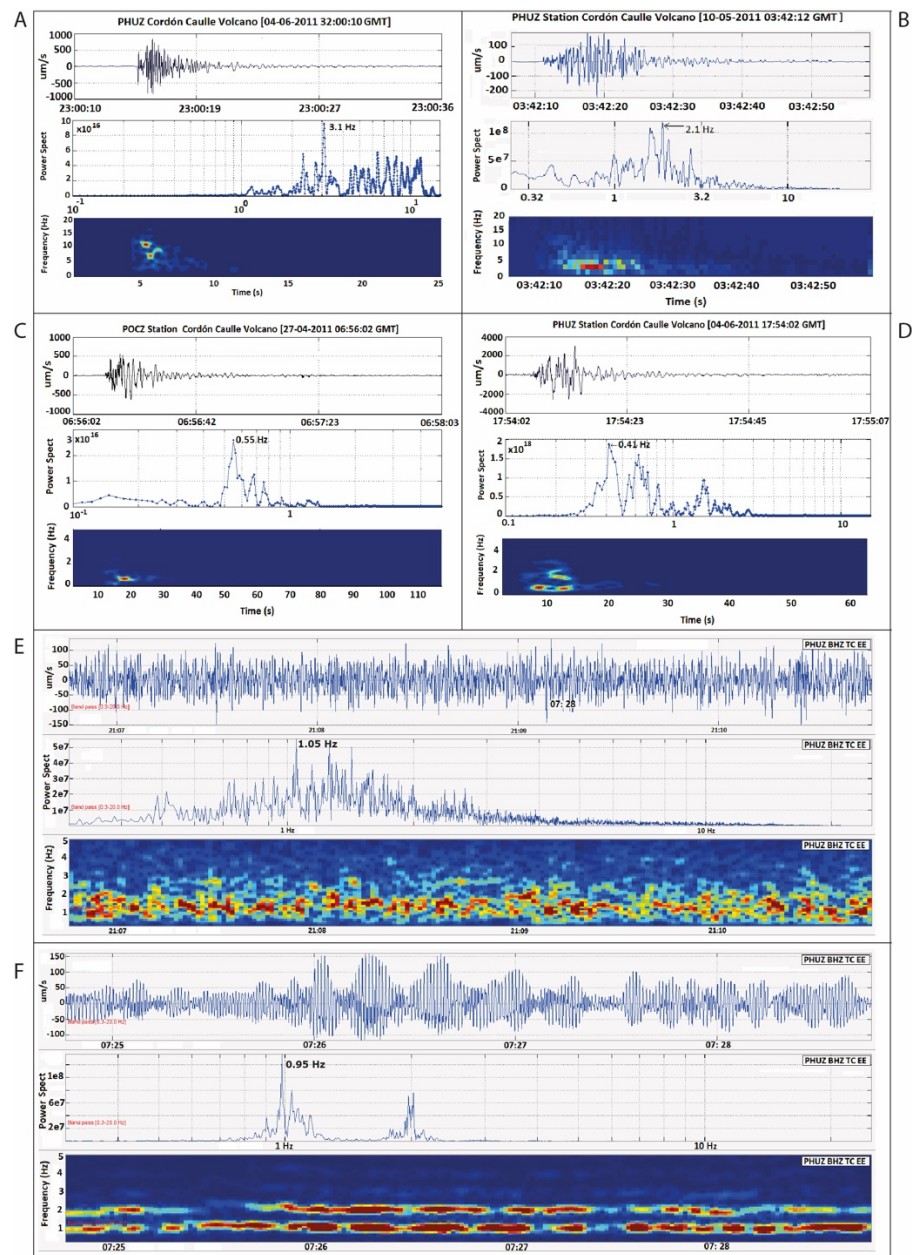

**Figure 2. Selected examples of different seismic signals recorded at station PHU (with the exception of C recorded at station POC). Each sub-figure has three panels; upper panel shows velocity seismograms in the vertical component, middle panel shows power spectra of the signal in a frequency range commonly between 0.1 and 10 Hz and lower panel shows a spectrogram of the frequency content in time, with more reddish colors indicating high energy concentration. A) Volcano-tectonic (VT) event. B) Long period (LP) event. C) Hybrid (HB) event. D) Very long period (VLP) event. E) Spasmodic tremor recorded during the explosive phase of the eruption. F) Quasi-harmonic tremor recorded during the effusive phase of the eruption.**

Local Magnitudes ($M_L$) were computed for VT and HB based on Hutton and Boore (1987). This first-level processing generated a database consisting of a total number of 32,850 earthquakes recorded between December 2010 and December 2011. We also used wave polarization method (Matsumura, 1981) to corroborate the existence of body waves in representative seismic traces (>10 $\mu m/s$) of HB, LP, VLP, and TR events. The polarization method helped to estimate locations (azimuth) for case of TR seismic activity (Bertin et al., 2015). To quantify the size of TR events (spasmodic and quasi-harmonic), the reduced displacement ($D_R$) was used (Aki, 1981). Finally, preliminary hypocenter location of 1750 well-recorded events were calculated using HYPO71 program (Lee and Valdés,1985).

### 3.3 Second level processing of the seismic data

A second level of processing was implemented with the aim to refine the location and characterization of seismicity, including the following procedures:

### 3.3.1 1D seismic velocity model and hypocenter relocation

We performed a relocation process of the recorded events using the Join Hypocenter Determination (JHD) algorithm (Crosson, 1976; Ellsworth, 1977; Thurber, 1983). For this, the local one-dimensional (1D) crustal velocity model was refined from the regional model of Bohm et al. (2002) using VELEST (Kissling et al., 1994) and considering only 425 high quality events (8 or more phases; horizontal errors<0.5 km, vertical errors<1 km; RMS solutions<0.15 s; gap <180°) from the 1750 preliminary located earthquakes. The reference level of the 1D model was established at 1.5 km above sea level. According to the methodology proposed by Kissling et al. (1994), we proceeded to verify the results, fixing the 1D velocity model and time delays to each station, giving only degrees of freedom at 100 hypocenters, restoring most of the best locations. Finally, considering the new input velocity model, we proceed to relocate all the original 1750 events.

### 3.3.2 Determination of focal mechanisms

Focal mechanisms of relatively large events ($M_L$>3.0) were computed using FOCMEC (Snoke et al., 1984) included into SEISAN program (10.1version; Havskov and Ottemoller, 1999) and considering first motion for events with 7 or more impulsive polarities recorded by the seismic network. To improve the solutions, the nodal planes were selected using the S/P ratio amplitudes (Hardebeck and Shearer, 2003). For some particular cases of large VT and HB events, we also computed the whole seismic moment tensor using waveform inversions with ISOLA software (Sokos and Zahradnik, 2008). To eliminate noise, the seismic recorded were filtered to frequencies <0.25 Hz before performing point-source deconvolution iterations. The calculation of the Green Functions associated to the total wave field generated by the elastic behavior of the crust was developed following the methodology proposed by Bouchon (1981). The moment tensor is calculated using the least squares

method. The time of origin and the spatial position of the point source (centroid) were searched around the relocated hypocenter.

### 3.3.3 Temporal variation of *b-value*

We investigate the temporal evolution of seismicity by means of the frequency-magnitude relationship as parameterized by
the *b-value*, which is the slope on the Gutenberg and Richter (1944) earthquake-size distribution. We applied the b-positive
method of van der Elst (2021) that considers only the positive magnitude differences of successive seismic events instead of
their actual magnitudes. This allows to overcome the potential bias in the computation of time series of b-value that arises
when the magnitude of completeness $M_c$ of the seismic catalog change with time, as is our case. After replacing in the cataolog
the magnitude of each event by the magnitude differences with the preceding event (in cases that this difference is positive),
we then applied the classical maximum likelihood method (Aki, 1965; Hamilton, 1967) to compute the b-value of sample of
events in a given time window. We worked with the 1750 preliminary located events that have reliable estimated magnitudes.
We used correlative and overlapping time windows defined by the occurrence of 100 events with an overlap between them of
90 events. For each window we compute one *b-value* and its error (as estimated using Aki, 1965).

### 4 Chronology of 2011 Eruption

We describe the temporal evolution of PCCVC before and after the eruption´s onset of June 4[th], 2011 (Fig. 3). For this, we
established different phases considering changes in time series of parameters derived from the first level processing of the
recorded data. This is complemented with hypocenter relocations, focal mechanisms and *b-value* time series contributed by
the second level processing and a revision of relevant published results on the time evolution of PCCVC between 2003 and
the end of 2011. The latter is mostly based on observed changes of eruptive style after eruption onset and published analyses
of InSAR and GNSS data. Time-series of several parameters and definition of the 7 phases comprising our analysis are shown
in Fig. 3A-F. Fig. 4 shows an enlargement of the time window between May 21 and June 15 for the most significant parameters.
The spatio-temporal progression of relocated seismicity and computed focal mechanisms is presented in Figs. 5 to 8. Based on
this information we describe the following chronology.

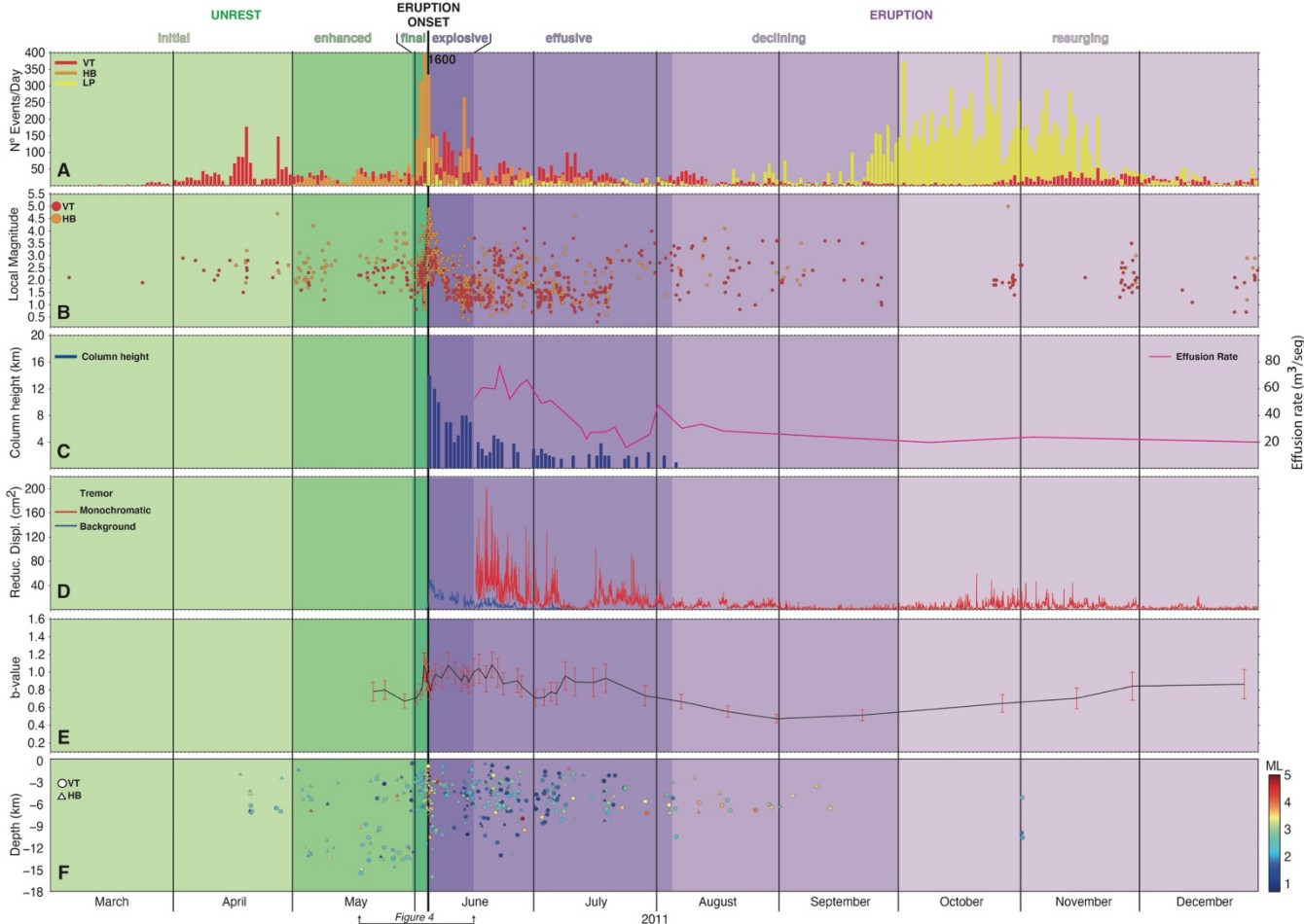

**Figure 3. Time series of relevant parameters before (March) and after (December) the 2011 PCCVC eruption (June 4th). A) Daily count of VT (red), HB (orange) and LP (yellow) seismic events recorded at station PHU (see Fig. 1). B) Local magnitudes from 1750 well-located VT (red) and HB (orange) earthquakes. C) Height of the pyroclastic column (blue) and effusion rate (red, after Bertin et al., 2015). D) Reduced displacement of the background tremor (blue) accompanying the Explosive Phase and the monochromatic tremor (red) dominating the Effusive Phase. E) *b-value* (grey line) with error bars (red). F) Hypocenter depths and magnitudes for the best located 425 VT (circles) and HB (triangles) earthquakes. At the bottom of the figure we mark the time window enlarged in Fig. 4. Colour panels and text on top of the figure define the different phases described in the main text.**

## 4.1 Pre-Unrest Phase (2003 to November 2010)

Interferometric Synthetic Aperture Radar (InSAR) images obtained before the eruption and summarized by Delgado (2021) indicate at least three distinct episodes of surface uplift between 2003 and the end of 2010, at rates that increased from 3-4 mm/yr (2003-2008) to ca. 30 mm/yr (2008-2010).

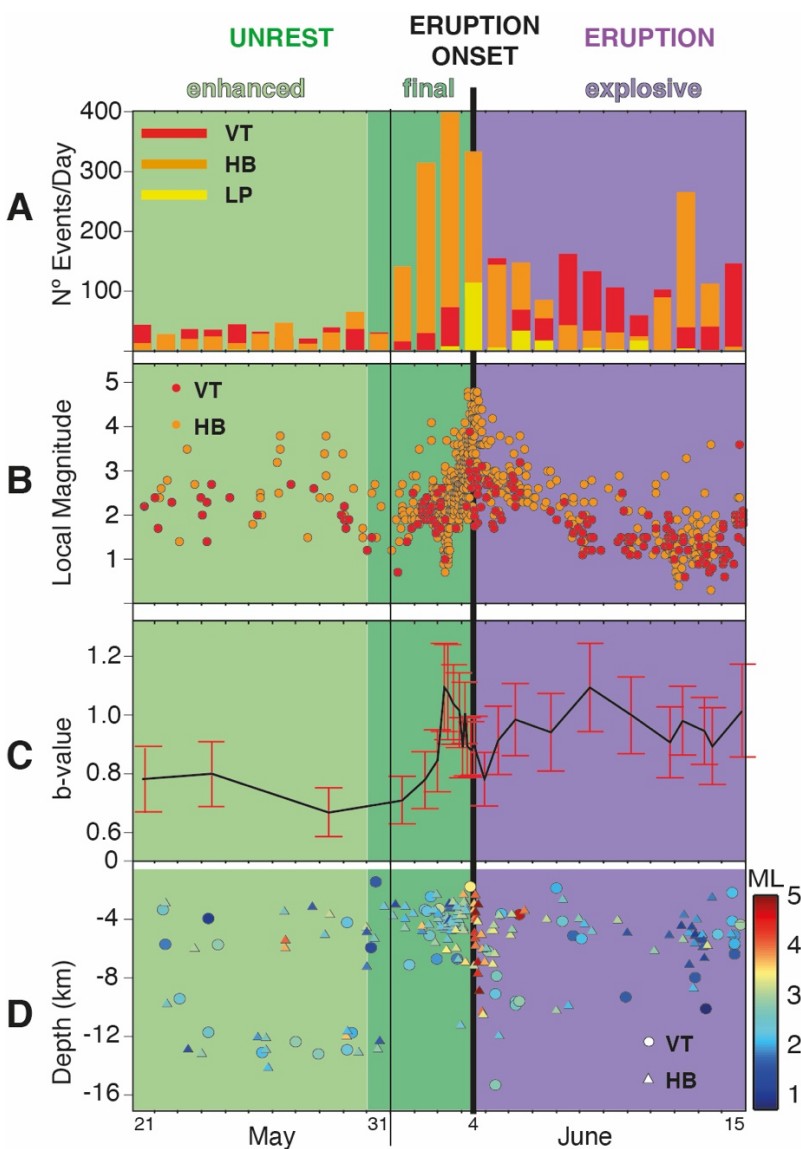

**Figure 4. Enlargement of the time window between May 21 and June 15 for number of events per day (A), local magnitude (B), b-value (C) and location depth (D). Symbols and color coding are as in Fig. 3.**

Sources for these uplift episodes have been modelled as inflating sill and points located between 4 and 9 km depth below Cordón Caulle graben and Cordillera Nevada volcano (Fig. 5) and have been interpreted as events of mafic magma injection inside a crystal mush reservoir. The last inflation episode was accompanied by clusters of moderate magnitude ($M_L$ 1.5-4.0) VT earthquakes identified by the regional OVDAS network and located up to 10 km depth below Cordillera Nevada. After the

Mw8.8 Maule earthquake of 27 February 2010, Jay et al. (2014) identified a shallow (1.7 km depth beneath the surface) inflation event below Cordillera Nevada, likely associated with the activation of the hydrothermal system.

230

### 4.2 Unrest (December 2010 to June 4, 2011)

### 4.2.1 Initial Phase (December 2010 – April 25, 2011)

The regional network of OVDAS recognized first signs of unrest by the end of 2010 beneath the Cordillera Nevada, observing a sporadic record of low frequency events classified as VLP and HB. The subsequent installation of near field stations at the flanks of PCCVC between January and February 2011, allowed capturing details of seismicity through time. Until the first half of March, some VT earthquakes per day were recorded with $M_L<3$. Although some of these events (mostly at the beginning of the sequence) can reach up to 10-15 km, most of them were concentrated at relatively shallow depth (<6-8 km beneath the surface), above the inflation sources derived from InSAR during the pre-unrest phase. By the end of March, the number of VT earthquakes increased. This tendency continued in the first half of April when episodes of this kind of seismicity (mixed with some LP and HB events) lasted several days, registering up to 180 events/day with magnitudes up to $M_L=4$. These events were mostly located underneath Cordillera Nevada at depths <7 km, although part of them occurred underneath the south-western branch of Cordón Caulle relatively close to the future eruptive vent. After a cluster of seismicity on April 18th at shallow depths (<4 km beneath the surface) below Cordillera Nevada, a period of relative seismic calm closed this phase.

### 4.2.2 Enhanced Unrest Phase (April 26 - May 30)

A reactivation of mixed VT and HB seismicity during the last week of April culminated the 27th with a very energetic event ($M_L=4.7$) classified as HB. This event was located near the intersection of Cordón Caulle northeastern branch with the rim of Cordillera Nevada at 8 km depth, i.e. in the range of the deepest pre-2010 inflating InSAR source. It was followed by a cluster of decreasing number of events. During the first half of May, the number of VT kept constant in ca. 30-40 events/day with magnitudes up to $M_L=3.5$, whereas HB events start to be more notable and reached larger magnitudes ($M_L=3.5-4$). Epicenters of these events tend to cluster at the same two locus detected in the previous phase, but we note a tendency to cover a larger depth range from 12 km below Cordillera Nevada to <5 km underneath Cordón Caulle (Fig. 5). After a relative seismic quiescence at the mid of May, the second half of this month was characterized by a relatively constant rate of seismicity of mixed VT and HB on the order of 35-45 events of each type per day, but maximum magnitudes were gradually increasing ($M_L>4$) and locations were concentrating toward the central and eastern side of the graben with shallower depths (Fig.5).

255

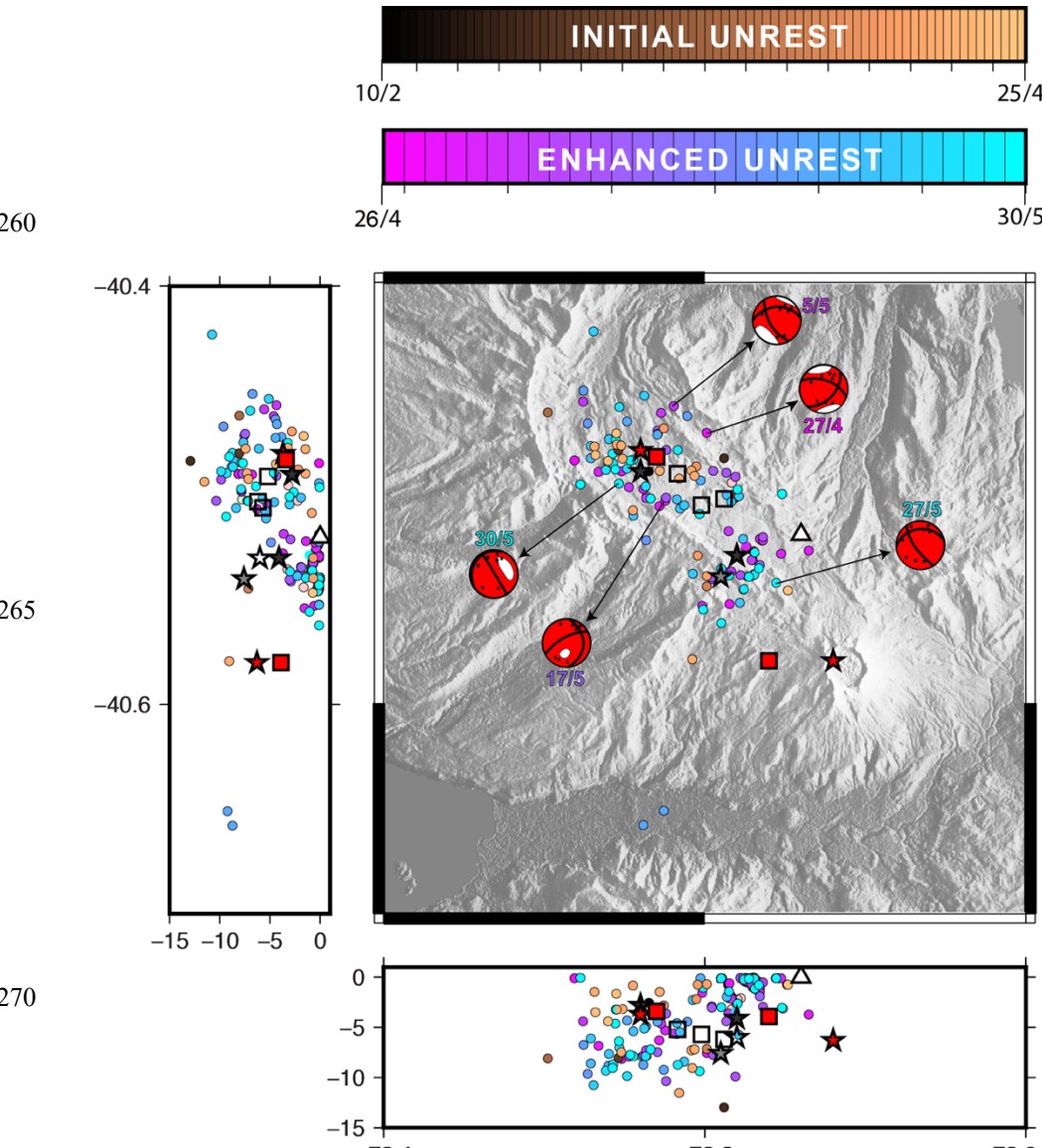

**Figure 5. Relocations of seismicity recorded before eruption onset including the Initial and Enhanced Phases. The location of hypocenters are indicated with circles with colors representing time in days (see color codes at the top of the figure with numbers showing day/month during 2011). Red/white beachballs with numbers are focal mechanisms with date of occurrence calculated by inversion using ISOLA software. The black squares located inside the inversions represent the polarities, complementing the solutions of focal mechanisms inversion. The black arrows connect the epicenters with the focal mechanisms solution. White triangle is the eruptive vent. Location of deformation sources modelled by Jay et al. (2014, stars) and Wendt et al. (2016, squares) are either transparent (if they are inactive at the time of the Initial and Enhaced Unrest phases) or red if they were active during the eruption onset. The negative numbers in the profiles represent depth in km from the reference level of the 1D seismic velocity model (at 1500 m.a.s.l.).**

The larger rate of seismicity at the end of this phase compared with previous months allows computing the first two points of the *b-value* time-series (Fig. 3E and 4C), which show relatively low values around 0.8 that could be interpreted as an indication of a highly stressed system. This interpretation can be confirmed by focal mechanisms of five large HB events recorded in this phase that were computed using full waveform inversion (Supplementary Material, Figure S2 and Table S1) and were complemented by first polarities (Fig. 5). Focal mechanisms are dominated by a non-double-couple (NDC) component, with percentages of double couple as low as 40% (Supplementary Material, Table S1). The NDC is formed by an undetermined percentage of an expansive isotropic component (ISO, also known as volumetric, e.g. Křížová et al., 2013) and a compensated linear vector dipole (CLVD) component. This combination has been commonly interpreted in other volcanic areas as caused by pressurized magma mobilization and intrusion (Aki, 1984; Julian and Sipkin, 1985; Dreger et al., 2000; Sarao et al., 2001; Templeton and Dreger 2006).

### 4.2.3 Final Unrest Phase (31 May – 4 June)

The week before the eruption was characterized by an increasing number of VT, LP and mainly HB events at depths commonly shallower than 5 km (Fig. 3). The number of events and their magnitudes gradually increased with time. Hypocentres were no longer associated to Cordillera Nevada and form a somehow diffuse grouping along the central part of Cordón Caulle graben and a sharper NE-SW alignment at the western flank of Puyehue volcano, likely along an eastern branch of LOFZ (Fig. 6). The future erupting vent seems to be located at the north-eastern intersection of these two groups of events. On June 2[th], and considering the notable intensification of seismicity, OVDAS changed the volcanic alert to yellow. In the following days, seismicity rate and magnitudes increased recording up to 1600 HB events the day before the eruption and more than 200 events/hour some hours before the eruption (Supplementary Material Figure S1).

At least 25% of these events presented relatively large magnitudes between $M_L$ 3.0 and 4.9. The cumulative total energy (computed using Boatwright, 1980) of events recorded before the eruption was $1.65 \times 10^{20}$ ergs, representing an accumulated seismic moment equivalent to a Mw5.6 earthquake. The intense final seismic swarm was characterized by an increase of *b-value* to a maximum of 1.1 and then a rapid decrease to 0.8 just before the eruption onset. Attending to the great intensity of recorded seismicity, OVDAS decided to warn a red alert on the morning of June 4[th], activating the evacuation of the population within a radius of 25 km around the volcano.

### 4.3 Eruption stage (4 June 2011 to January 2012)

### 4.3.1 Onset (4 June, 18:45 UTC (14:45 hours local time))

Culminating the intense seismic activity at the end of the Final Unrest Phase, a Plinian VEI 4-5 explosive eruption took place, opening a new vent near the intersection of the northeastern branch of Cordón Caulle graben with the trace of LOFZ. A sustained and vigorous eruptive column was fed from the vent at initial rates near $10^7$ kg/s (Pistolesi et al., 2015) and rose to an altitude of 10-12 km in less than one hour.

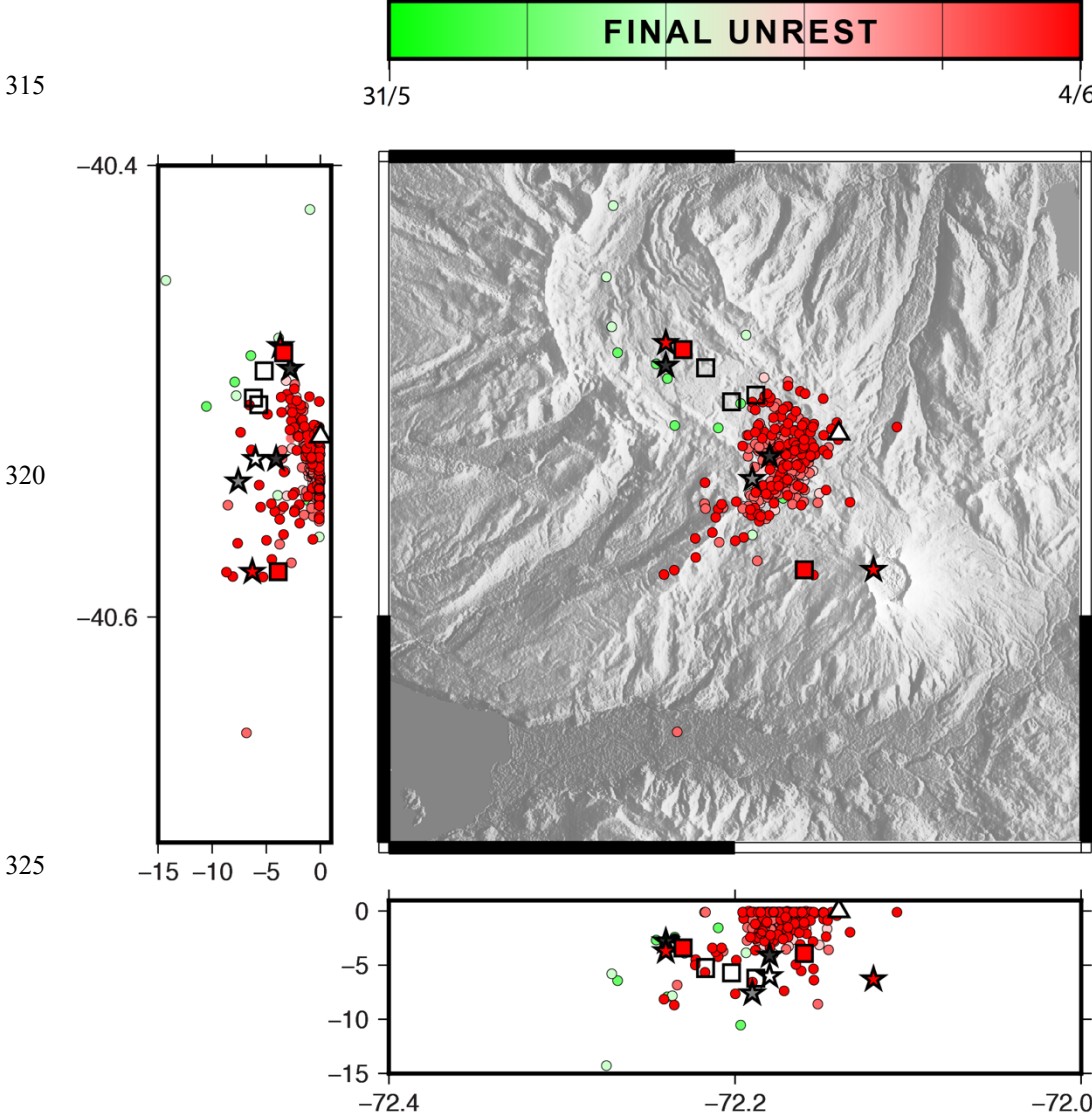

**Figure 6. Relocations of seismicity recorded before eruption onset during the Final Unrest phase. See caption of Fig. 5 for an explanation of symbols and colors.**

### 4.3.2 Explosive Phase (June 4 – June 14)

The paroxysmal starting phase lasted 2-3 days, with a pyroclastic column that reached up to 14 km height and at least five episodes of partial column collapse that generated pyroclastic density currents mainly heading north (Pistolesi et al., 2015). Over the following week, column height oscillated in response to frequent variable-sized explosions and steadily reduced to 4-8 km height (Fig. 3C). These changes were coeval with a period that included both sustained tephra emission and the ballistic ejection of large bombs. This coincided with localized surface uplift underneath the vent, which was interpreted as produced by a shallow laccolith intrusion (Castro et al., 2016). On 12-13 June, pulsating pyroclastic jets were accompanied by intense ballistic activity that marked the last major events closing this phase (Pistolesi et al., 2015; Castro et al., 2016; Shipper et al., 2021).

An interferogram formed by SAR scenes acquired on 8 May and 7 June 2011, captured the surface deformation produced by the first 3 days of the eruption (Jay et al., 2014; Wendt et al., 2016; Euillades et al., 2017; Delgado, 2021; Novoa et al., 2022). This InSAR image is characterized by two lobes of metric-scale subsidence joined by an area elongated along Cordón Caulle graben. One lobe is collocated with the rim of Cordillera Nevada and shows up to 1.2 m of LOS subsidence, whereas the second less prominent and less coherent lobe lies southward of the new vent. The source of deformation has been modeled by two deflating point sources at depths of 3.8 and 6.1 km (Jay et al., 2014, Wendt et al., 2016). The large northwestern deflating source coincides with one of the inflation sources recognized in the pre-unrest phase, whereas the smaller southeastern point is not related to older sources. Wendt et al. (2016) demonstrated that the northwestern deformation signal could also be modeled by a closing NW-oriented southward-dipping dike, with a significant component of left-lateral strike slip movement, connecting the rim of Cordillera Nevada with the active erupting vent.

After the eruption onset, we observe (Fig. 3A) a sharp decrease in the number of HB events and an increase of VTs compared with the Final Unrest Phase. Hypocenter locations for the first 2-3 days of the Explosive Phase (Fig. 7) mostly coincide with those observed just before the beginning of the eruption, although they cover a larger area along the Cordón Caulle graben and the western flank of Puyehue volcano. The latter roughly coincides with the smaller deflation point recognized by Wendt et al. (2016), whereas the larger co-eruptive deflation sources underneath Cordillera Nevada (Wendt et al., 2016; Jay et al., 2014) remains aseismic during these first couple of days. This seismicity was accompanied with a spasmodic tremor, which became the dominant component of the seismic signal since June 6 (Bertin et al., 2015). The multi-frequency character (0.5-4 Hz) and constant oscillations in the amplitude of the tremor (as measured by the $D_R$, Fig. 3D) evidenced a possible combination of sources as explosions, ash and gas output, and several pyroclastic flows. The location of the seismic activity during the pre-unrest phase coinciding with the position of the largest co-eruptive deflation source (Jay et al., 2014; Wendt et al., 2017) suggest that the main magmatic reservoir feeding the eruption was located underneath Cordillera Nevada between 6-7.5 km.

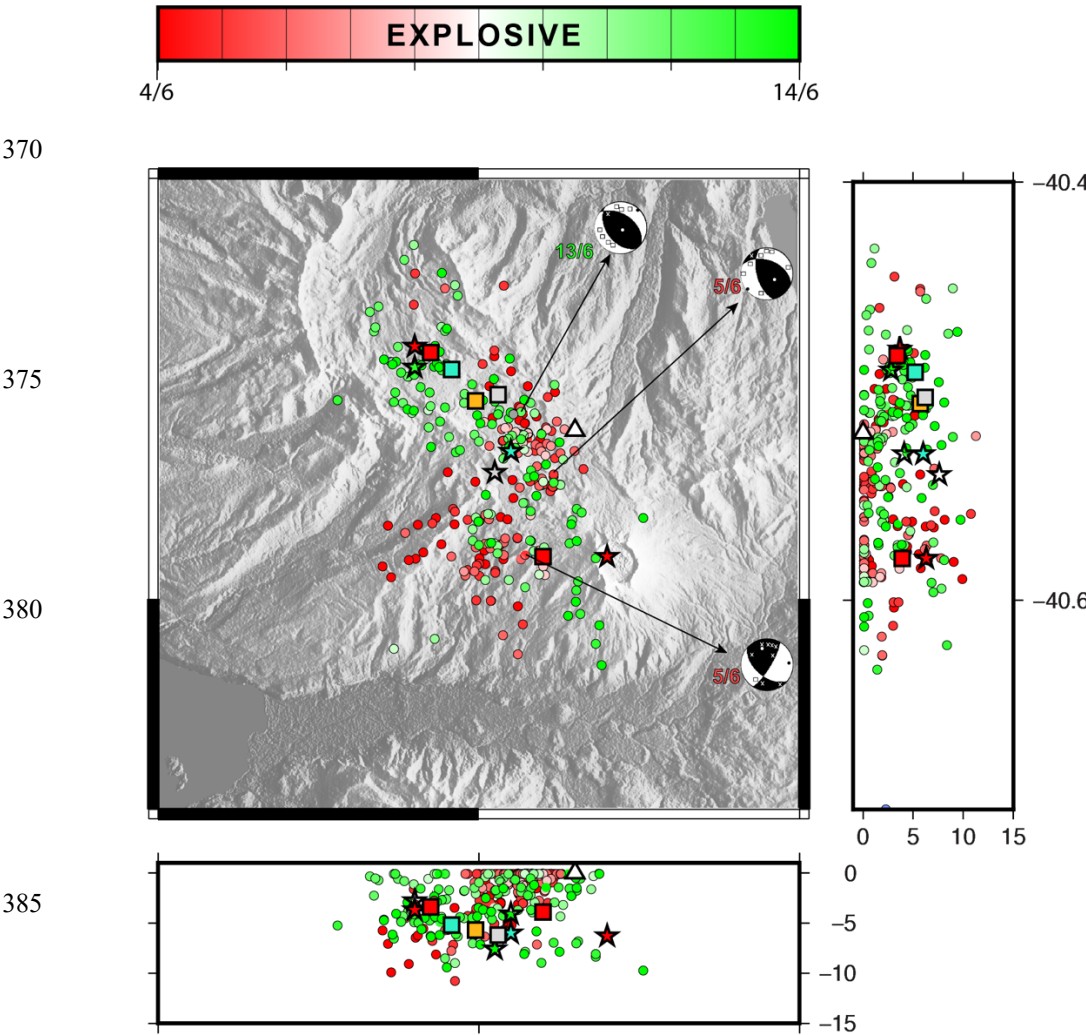

**Figure 7. Relocations of seismicity recorded after eruption onset during the Explosive phase. See caption of Fig. 5 for an explanation of symbols and colors. In this case, focal mechanisms (white/black beachballs) were calculated by first arrivals, with squares and crosses representing negative and positive polarities**

Further changes in the character of the Explosive Phase after June 8-9, when the height of the pyroclastic column diminishes and vent activity started to be dominated by ballistic bomb explosions, was marked by a continued decrease in the rate and magnitudes of HB and VT events (Fig. 3), and an expansion of the area illuminated by hypocenters back to Cordillera Nevada. A NS alignment of seismicity crossing the western flank of Puyehue volcano and likely associated to the LOFZ was also activated in this period.

On 12-13 June, pulsating pyroclastic jets were accompanied by an oscillating tremor signal, increased number of HB seismicity and vigorous explosions that cast large ballistic bombs to distances up to 2.5 km from the vent (Castro et al., 2013). By the end of this phase, the rate of seismicity decreased, the amplitude of the spasmodic tremor became minimal and eruptive column height waned to less than 3 km.

We computed focal mechanisms using first arrivals analysis of three large earthquakes of this phase (Fig. 7). The two of them that are relatively close to the eruption vent show a reverse mechanism with nodal planes parallel to Cordón Caulle graben. We note that these focal mechanisms could indicate the prevalence of compressive and/or pressurized conditions near the vent during this phase, as can be also expected due to the intrusion of magma at high rate to form the laccolith inferred by Castro et al. (2016). The other focal mechanism computed for this phase corresponds to an earthquake belonging to the cluster on the
western flank of the Puyehue volcano and shows a strike-slip mechanism with possible dextral motion along a nodal plane roughly parallel to the LOFZ.

The *b-value* time series (Fig. 3E and 4C) shows an increase of this parameter during this phase with some oscillations around values 1 to 1.2. This is consistent with a gradual decrease of the stresses supported by the system as it becomes relaxed by the evacuation of gas and ash throughout the vent.

**4.3.3 Effusive Phase (15 June– 6 August)**

Since 15 June, the entire seismic network began to record energetic low-frequency pulses constitutive of a quasi-harmonic tremor (Bertin et al., 2015). The $D_R$ of this signal was 10-15 times larger than the spasmodic tremor of the explosive phase (Fig. 3D), indicating a significant change in the behavior of the eruption. A lava flow was confirmed on 17 June by satellite images and later on 20 June by an overflight (Bertin et al., 2015). As summarized by Bertin et al. (2015), tremor particle
motions showed a P-wave nature with incidence angles close to sub-horizontal, pointing out to a shallow source and with virtually no changes of the dominant frequency of the tremor during the entire effusive phase. These tremor oscillation features suggested the fast movement of magma through the eruptive vent. We complemented the work of Bertin et al. (2015) performing a polarization analysis of the tremor signal at three stations (PHU, QIR and ANT, see Fig. 1) and confirming a dominant polarization azimuth aligned with the direction between each station and the eruptive vent (see Supplementary Figure
S5). This reinforce the idea that the quasi-harmonic tremor is sourced in the discharge of lava from the vent.

A significant temporal correlation between $D_R$ of the tremor and discharge rates of lava flow estimated by satellite images (Fig. 3C and D), confirm this causal relation (Bertin et al., 2015). Discharge rates 8 days after the start of the effusive phase were as high as 70 m³/s, whereas the average eruption rate during this phase was 50 m³/s (Bertin et al., 2015). Notably, the
pyroclastic column was intermittently active over this time period reaching altitudes smaller than 3-4 km, which has been an

argument to describe this phase more like a hybrid explosive-effusive eruption and not necessarily as a purely effusive phase (e.g. Schipper et al., 2021).

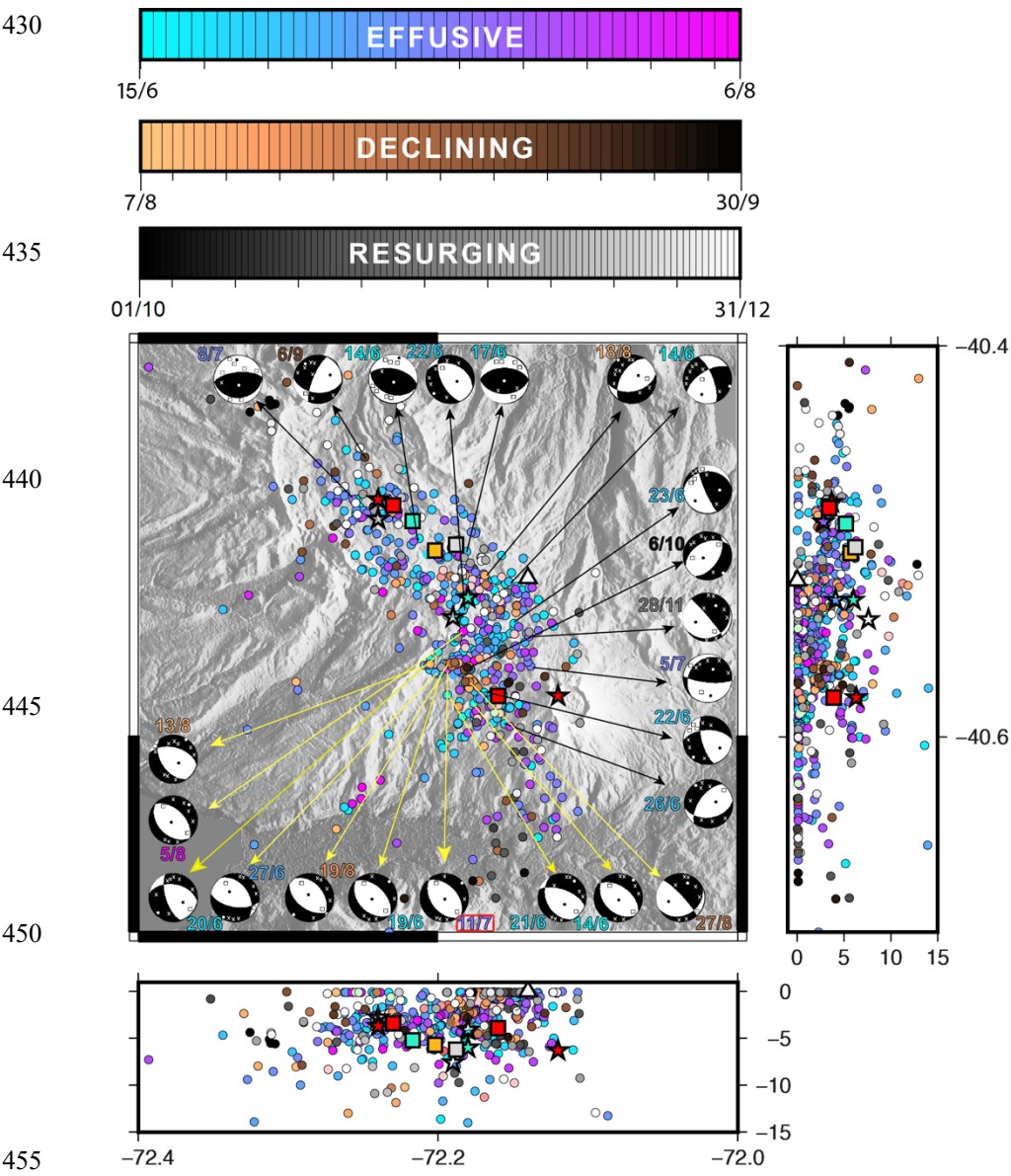

**Figure 8. Relocations of seismicity recorded after eruption onset during the Effusive, Declining and Resurging Phases. See caption of Figs. 5 and 7 for an explanation of symbols and colors.**

The initiation of this phase was accompanied by a decrease in the seismicity rate of VT and HB events, although the magnitude of these seismicity was somehow higher than during the end of the explosive phase. This implies a reduction of the *b-value* during the second half of June, suggesting a transient pressurization of the system. During the first half of July, the daily
number of seismic events kept relatively constant showing a prevalence of VT with respect to HB and LP. As the magnitude of these VT was relatively low, this implies a further increase of *b-value* that was stabilized around a value of one by mid-July. Simultaneously, the effusion rate gradually decreased together with the $D_R$ of the quasi-harmonic tremor. The last 2-3 weeks of the effusive phase were marked by a decrease of the seismicity rate, with relatively large magnitudes ($M_L$:2-4) mostly for VT earthquakes, which means a further decrease of the *b-value*. The effusion rate reached a peak of ca. 45 m$^3$/s the first
470 days of August and renewed energy of the tremor decline to a minimum at the end of the phase simultaneously with the disappearance of the pyroclastic column.

The main area occupied by hypocenters of events recorded during this phase (Fig. 8) does not differ markedly with respect of those of the Explosive Phase. However, we note that seismicity tends to expand outside the graben, mostly to the southern
flank of the complex, likely along the LOFZ. In addition, some of the events recorded in this phase reached depths of 10 to 15 km and are therefore much deeper than in the Explosive Phase. The areal and depth expansion of seismicity is consistent with a larger depth of the deflation source that explains the less prominent but more regional surface subsidence observed by InSAR during this phase. Combining SAR scenes for 7 June and 7 July, an interferogram presented by Jay et al. (2014) and Wendt et al. (2016) covering the waning stage of the explosive phase and the initiation of the effusive phase, was modeled with a single
deflating point source located 5-6 km below the active vent (Fig. 8). Delgado et al. (2019) presented an alternative solution with a deflating prolate spheroid at 5.2 km depth oriented along the Cordón Caulle graben that gives a better fit to the observed surface deformation for this interferogram and for those of coming months. Continuous GNSS observations presented by Wendt et al. (2016) are well-explained by their modeled point source, although a significant misfit in the vertical GNSS component in the southern flank of Puyehue volcano likely points to a localized movement of LOFZ that is consistent with the
observed concentration of seismicity.

Focal mechanisms computed for events of this phase and some occurring in the following Declining Phase highlight three main sources acting simultaneously. One of them is observed at the southern limit of Cordón Caulle graben, where a group of events exhibits a common focal mechanism of normal faulting with a dominant NW strike that is sub-parallel to the graben
(Fig. 8; yellow arrows). This is consistent with the ongoing relaxation of the system and deflation of the InSAR-modelled source of surface deformation (red square in Fig. 8 at the western side of Puyehue volcano). A second group of events located in the northern branch of Cordón Caulle and Cordillera Nevada is marked by a mix of normal and reverse focal mechanisms. The latter is hard to explain but it could be related to a continuous feed of magma toward the growing laccolith underneath the

eruptive vent. A third group of events with dominantly strike-slip focal mechanisms, is recognized at the eastern portion of the graben near the intersection with LOFZ, suggesting an activation of this structure.

### 4.3.4 Declining Phase (7 August – 30 September)

During this period, a small diffuse eruptive column was observed, discharge rates of lava kept constant at relatively low values (20-30 m$^3$/s) and $D_R$ of the tremor was minimal. Seismic activity remained at low levels and was dominated by sporadic VT events of intermediate magnitudes ($M_L$:2-4), located along the graben and to the south of the vent, likely along the LOFZ (Fig. 8). In accordance with the stability of seismic activity, the estimated *b-value* remains relatively constant at levels similar to the Enhanced Unrest Phase. During the second half of September, the number of LP events increased from some tens to hundreds of events per day, suggesting a new shallow intrusion in the eruptive process . InSAR images and GNSS data during this phase still show some subsidence but modeled sources (Jay et al., 2014; Wendt et al., 2016; Delgado et al., 2019) comprise lower volumes compared with previous phases.

### 4.3.5 Resurging Phase (1 October 2011– January 2012)

A rising number of LP seismicity during the second half of September anticipated a reactivation of the volcanic activity. This was associated with an increase in the amplitude of the quasi-harmonic tremor and a new emission of ash, explosions and effusion of lavas. The number of LPs and $D_R$ of the tremor then decreased during November and December, whereas clusters of VT (and minor HB) started dominating the seismicity record. Hypocenters associated to these clusters show a further expansion of seismicity outside the northwestern border of Cordillera Nevada caldera and mostly southward of Puyehue volcano at shallow depths (<5 km) along the main trace of LOFZ (Fig. 8). The occurrence of these clusters is associated to a gradual increase of parameter *b* to a value larger than one by the end of 2011, suggesting a further relaxation of stresses. This is consistent with a continued subsidence observed by InSAR until February 2012 (Jay et al., 2014).

## 5 Discussion

Here we attempt to highlight the main processes likely occurring inside the magmatic plumbing system of PCCVC before and after the eruption onset of June 4[th] 2011. This is done considering main features of the recorded seismicity described above along with results and models proposed by previous authors for this eruption. The methodologies used allows us to discuss general conceptual models regarding destabilization of structurally-controlled acidic magmatic systems, the pass from unrest to eruption, changes in eruptive style and waning phases of silicic eruptions, with implications for monitoring and forecast of violent rhyolitic eruptions. We integrate most of our results and relevant constraints provided by previous authors in the

schematic representation of the magmatic plumbing system underneath PCCVC as shown in Fig 9. Based on this figure, we also attempt to delineate the main processes occurring during the unrest phase and the eruption onset in Fig. 10.

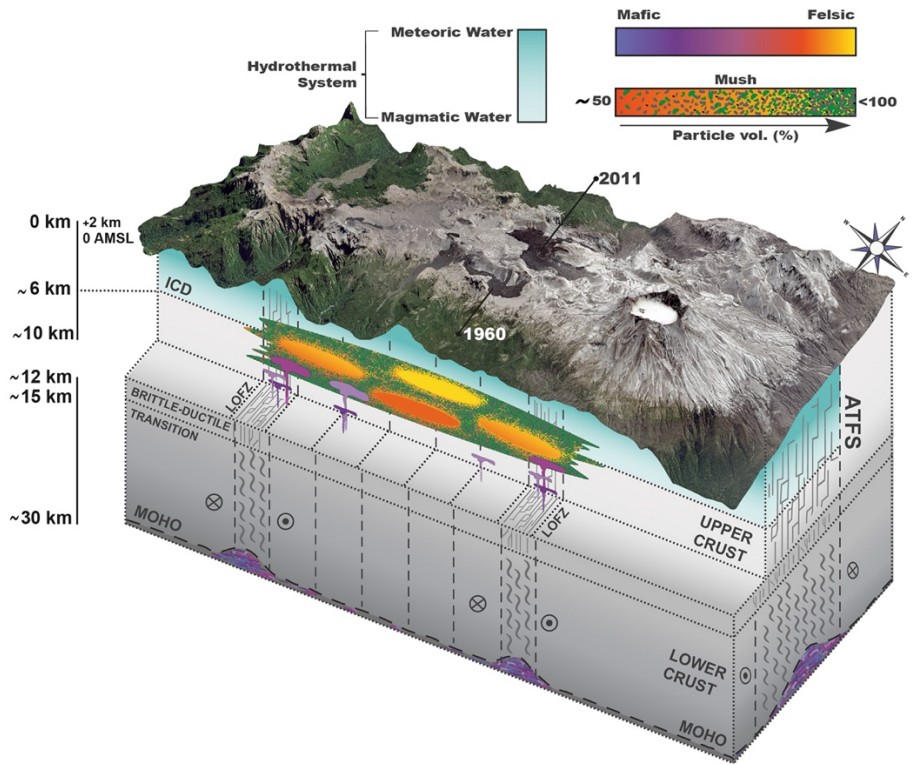


**Figure 9. Schematic representation of the crustal structure underneath Puyehue-Cordon Caulle Volcanic Complex. This includes the vertical stratification of the crust and the occurrence of main fault systems (Liquiñe-Ofqui Fault Zone LOFZ and Andean Transverse Fault System ATFS). See text for description and sources of information.**

## 5.1 Vertical zonation and upper crustal architecture of PCCVC

Average depth for sources of InSAR-observed surface deformation before the eruption and during its effusive phase are of the order of 4-8 km (Jay et al., 2014; Wendt et al., 2016; Delgado et al., 2019). They have been interpreted as marking the depth at which magma was injected years to months before the eruption at the base of the main silicic reservoir. This depth range roughly coincides with the depth of ~5-6 km below the surface for the Intra-Crustal Discontinuity (ICD) that separates light upper crust from dense mid-lower crust in the 3D forward gravity model of Tassara and Echaurren (2012). Oyarzún et al. (2022) have shown that the ICD likely constitute a fundamental discontinuity that controls the vertical zonation of the plumbing system underneath Nevados de Chillán Volcanic Complex and other petrologically well-studied Southern Andean volcanoes, separating a deep intermediate-to-basic reservoir from a shallow dacitic-to-rhyolitic evolved mush zone (Fig. 9).

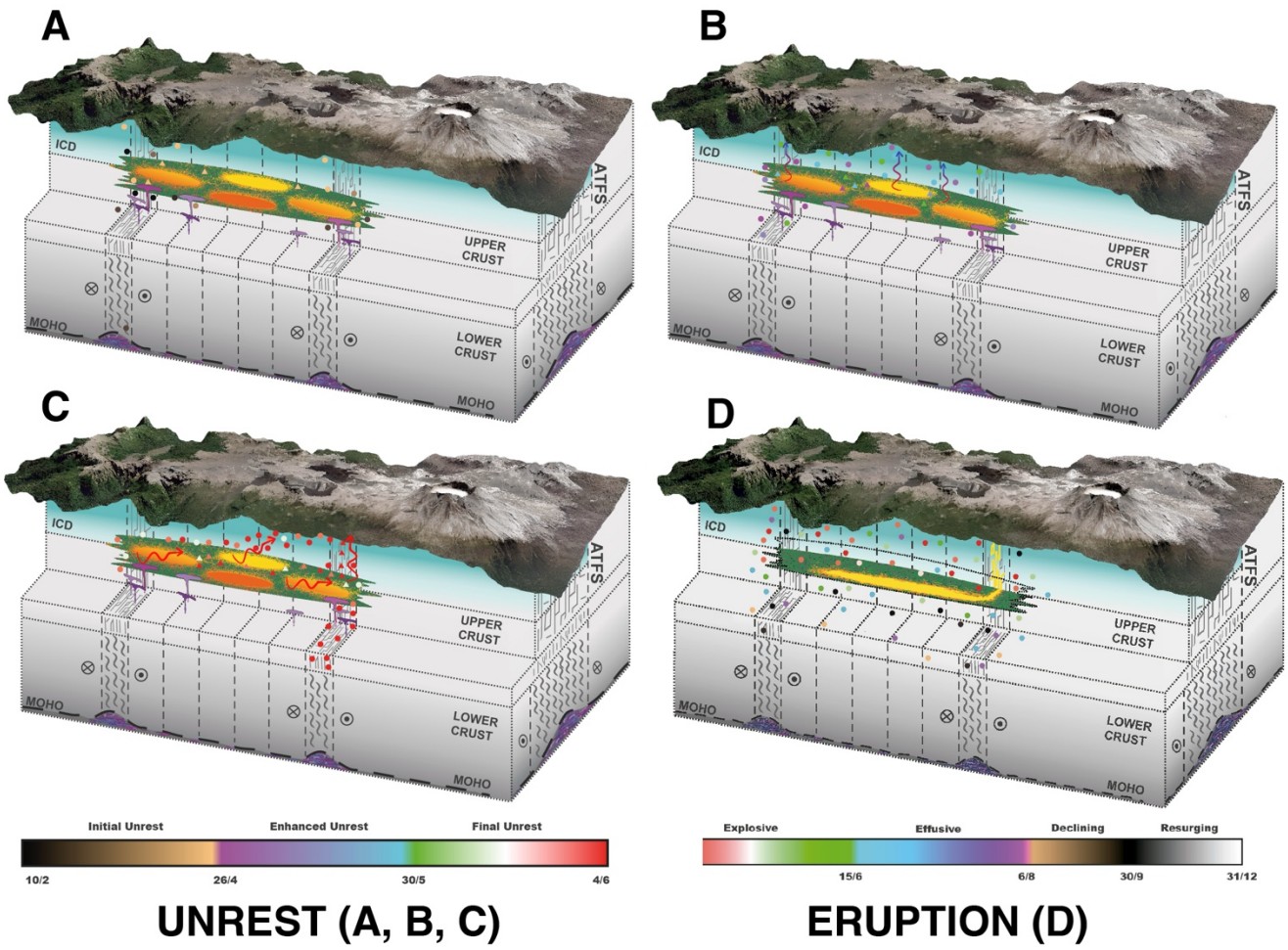

**Figure 10. Evolution of the magmatic plumbing system and seismicity before (A, B, C) and after (D) the eruption onset. Circles and triangles are respectively VT and HB seismic events. They are coloured as in Figs 5-8. Undulated arrows mark the pattern of fluid and magma migration.**


In support to this vertical zonation, the 1D seismic velocity model generated as a by-product during the relocation of seismicity (Supplementary Material, Figure S4) also shows a strong velocity contrast at 6 km below the surface, in coincidence with the depth of the ICD, the average depth of deformation sources and the maximum depth of acidic magma storage evacuated by the last three eruptions of Cordón Caulle (Castro et al., 2013). Interestingly, this depth also marks the bottom limit of seismicity

underneath Cordillera Nevada Caldera during the initial unrest phase (Figs. 3F, 5, 10A), suggesting that these first signs of destabilization took place at the boundary between a deep intermediate-to-mafic reservoir and the shallow silicic mush.

In addition to the vertical zonation of the magmatic system, our results also highlight the active nature of fault zones in the uppermost crust above the magmatic reservoir and the important structural control experienced by PCCVC before and mostly
after the onset of the eruption, as also suggested by previous authors (Alloway et al., 2015; Wendt et al., 2016; Delgado et al.,2019, 2021; Novoa et al., 2022). Seismicity recorded during the whole studied period is clearly aligned with the NW-oriented Cordón Caulle structural graben, which can be considered part of the Andean Transverse Faults Systems (ATFS in Fig. 10; e.g. Sanchez et al., 2013). Nodal planes of normal focal mechanisms recorded at the SE branch of the graben during the effusive and declining phases (Fig. 8) are parallel to Cordón Caulle orientation, further reaffirming the activity of this
structure as a normal fault associated to the relaxation of the upper crust after the main eruptive phases. The intense shallow seismicity recorded during the final unrest phase (Fig. 6) was concentrated near the eruption vent and seems to be related to the activation of the eastern branch of the LOFZ just before the eruption onset (Fig. 10C). However, the main NNE-oriented trace of this fault zone was activated at the end of the explosive phase and mostly during the effusive and declining phases as shown by the concentration of shallow seismicity to the south of Puyehue volcano (Figs. 8 and 10D). In contrast with focal
mechanisms of events along Cordón Caulle graben, those that could be associated with the LOFZ in Fig. 7 show a mixed behaviour between normal and strike-slip motions. This is compatible with a general relaxation of the upper crust after the main eruptive phases but also with the geologically-determined dextral strike-slip motion of the LOFZ (i.e. Cembrano and Lara, 2009).

## 5.2 Destabilization of a structurally-controlled rhyolitic system and the pass from unrest to eruption

As summarized by a recent Consensus Study Report of the US volcanological community (National Academies of Sciences, Engineering and Medicine, 2017; hereafter NASEM17), some of the most relevant open questions in volcano monitoring and eruption forecasting concern with the mechanical processes that initiate an eruption and how to recognize whether an instrumentally monitored unrest episode will end-up or not with an eruption. This concern is more critical for infrequent, less
studied and potentially most violent rhyolitic eruptions. Our results help shedding light into this gap of knowledge.

In particular, the observed temporal evolution of hypocentral depths (Fig. 3F) supports the "top-down" model of precursory volcanic seismicity proposed by Roman and Cashman (2018). Swarms of distant VT events, recorded several months to years before eruptions and located up to tens of kilometers away from future eruptive vents, are commonly thought to be the earliest seismic precursor of eruptions (White and McCausland, 2016). These kinds of events were recorded during the Pre-Unrest and
Initial Unrest Phases of Cordón Caulle located up to 10 km depth underneath Cordillera Nevada. Events recorded during the Enhanced and Final unrest phases show a larger depth range reaching up to 15 km depth and covering an area that was enlarging to the SE toward the final vent (Fig. 10B-C).

This behavior mimics the time-depth pattern of seismicity described by Roman and Cashman (2018) for well-monitored erupting volcanoes that have been in repose by decades and allows to apply their model to the case of the 2011 Cordón Caulle eruption. This model explains the processes driving unrest to eruption in three phases. The initial 'staging' phase can occur within years to months before the eruption and involves the movement of magma from mid to shallow crust, pressurizing the system and preparing the future conduit for the eruption. This phase could be completely aseismic if magma injection is slow and/or at small volumes, or can be accompanied by relatively deep seismicity. As described for our pre-unrest phase, the three InSAR-inferred episodes of magma injection between 2003 and 2009 (Jay et al., 2014; Delgado et al., 2019 and 2021), along with relatively large ($M_L$<4) VT earthquakes up to 10 km depth underneath Cordillera Nevada accompanying the last episode, are fully in line with the occurrence of the staging phase.

The second step of the "top-down" model of Roman and Cashman (2018), is the 'destabilization' phase when swarms of shallow VT seismicity are triggered by an over pressurization of the already staged upper-crustal magma body. The destabilization mechanism leading to over-pressurization can be magma vesiculation or second boiling caused by continued fractional crystallization (Stock et al., 2016), slow (aseismic) magma intrusion into the shallower reservoir, quiescent degassing (Girona et al., 2015) and/or the shallower intrusion of a small magma 'quanta' that precedes the bulk of magma finally evacuated (Scandone et al., 2007). Whatever the actual mechanism, general features of seismicity recorded during the Initial (Fig. 10A) and Enhanced (Fig. 10B) Unrest Phases of the 2011 Cordón Caulle eruption coincide with the destabilization stage of Roman and Cashman (2018). Clusters of shallow VT seismicity seem to be activated after signs of LP and HB events recognized by the regional network of OVDAS by the end of 2010, i.e., before the installation of the local seismic network. Low-frequency events with NDC mechanisms are considered a signal of magma and/or fluid mobilization at depth, likely associated in this case to the destabilization of the shallow reservoir that leads to pressurization of the upper crust above it. The relative coincidence between the location of VT clusters and the two main sources of pre-unrest surface deformation underneath Cordillera Nevada and Cordón Caulle (close to the final vent), indicates that initial destabilization can occur at separated locations inside the large NW-elongated silicic mush feeding the PCCVC (Alloway et al., 2015; Castro et al., 2013; Delgado, 2021) and seems to be related to regions that were already pressurized by past magma intrusion (Fig. 10B). Importantly, this final pressurization didn't produce significant surface deformation recognizable by InSAR, although a weak pre-eruptive deformation signal has been interpreted by Jay et al. (2014) as activity of the very shallow (<2 km) hydrothermal system. Another important observation is that VT swarms of the destabilization phase can be very energetic, counting up to some hundreds of events per day with magnitudes up to $M_L$:4-5, similarly as described by other erupting volcanoes (Roman and Cashman, 2018 and references therein).

The last stage of the top-down model is the 'tapping' phase exemplified in our case by the Enhanced and mostly Final Unrest (Fig. 10C) phases when seismicity starts to cover a larger depth range and mechanical connection is stablished between the main magmatic reservoir and the surface, finally triggering the eruption. Under the strong structural control of the magmatic

system of PCCVC and given the destabilization of at least two separated regions of the shallow silicic reservoir, this process shows a marked lateral migration of seismicity along the Cordón Caulle graben in addition to the enlarged vertical distribution of events. Actually, the deepening of seismic events that is typical of the 'tapping' phase of Roman and Cashman (2018) is well observed only underneath Cordillera Nevada, i.e., at the northwestern tip of the PCCVC about 12 km away from the erupting vent. However, seismicity clustered close to the vent was always shallower than the local pre-unrest inflation sources and become very shallow (<2-3 km) and strongly concentrated near the vent during the final unrest phase, i.e., the week before the eruption. These observations suggest that the 'tapping' phase of the 2011 Cordón Caulle eruption was able to open interconnections between the different active magmatic sources of the shallower silicic mush and with the surface utilizing the preferential path dictated by an intersection of pre-existing faults.

It is worth noting that a mix of VT and HB earthquakes began to occur during the enhanced unrest phase, suggesting that the final destabilization of the system and its tapping was not only associated to opening of fractures (VT events) but also to the mobilization of magma and/or fluids toward the surface throughout the opened structural paths (HB events). This process was accompanied by a further pressurization of the upper crust as suggested by expansive non-double couple (NDC) focal mechanism of large ($M_L$:3.6-4.7) HB earthquakes, which has been interpreted in other volcanic areas as caused by pressurized magma mobilization and intrusion (Aki 1984; Julian and Sipkin 1985; Dreger et al., 2000; Sarao et al., 2001; Templeton and Dreger 2006). Relatively low *b-values* computed during the enhanced unrest phase are in agreement with high pressures if we consider the inverse relationship that exist between differential stress and *b-value* (i.e., Schorlemmer et al., 2005; Scholz, 2015). Following the empirical relationship of Scholz (2015) a b-value around 0.8 for the enhanced unrest phase means a differential stress of 300-500 MPa. The gradual increase of b-value at the beginning of the final unrest phase to values near 1.2 suggest a reduction of the differential stress to less than 50 MPa (Scholz, 2015) that can be associated to a transient relaxation of the system as magma and fluids were able to open the conduit between the reservoir and the vent.. At the end of the 'tapping' phase, a final stage of run-up seismicity that is commonly observed right before eruptions (White and McCausland, 2016; Roman and Cashman, 2018) was largely dominated in this case by HB seismicity that become strongly concentrated at shallow depths (<2 km) and toward the intersection of Cordón Caulle graben with an eastern branch of the LOFZ, clearly pointing to the location of the future vent (Fig 10C). When the mechanical resistance of this structurally-damaged zone was finally surpassed, the conduit connecting the shallow silicic reservoir with the surface was established and the violent explosive eruption started (Fig. 10D).

**5.3 Changes in eruptive style and waning phases of a rhyolitic eruption**

Culminating the very intense seismic activity at the end of the final unrest phase, the eruptive vent was opened (Fig. 10D). The violent eruption onset was followed by three days of purely explosive activity associated to a 12-14 km high pyroclastic column that dispersed an unknown volume of gases and a tephra amounting ca. 1 km$^3$ of mostly rhyolitic ash (Pistolesi et al., 2015).

This paroxysmal explosive phase was characterized by a locally strong (up to 1.2-1.5 m) InSAR-observed surface subsidence that has been modelled by two deflationary sources roughly coinciding with the main pre-eruptive inflationary sources (Jay et al. 2014; Wendt et al., 2016; Delgado et al., 2019, 2021). In complement to these findings, our results indicate that the purely explosive phase of the eruption is likely characterized by a rapid relaxation of the structurally-controlled magmatic system, as shown by a notable decrease in the daily number of earthquakes (with VT dominating over HB) and their magnitudes. This is associated to a rapid increase of *b-value* right after the eruption and then fluctuations around a value of 1.2 that further reflects a reduction of stresses (Schoerlemmer et al., 2005; Scholz, 2015) affecting the system after the depressurization caused by the eruption onset. A marked depressurization is also compatible with the exponential decrease in the amplitude of the background spasmodic tremor that started to be registered right after the eruption onset (Bertin et al., 2015). The decrease of tremor´s amplitude is likely associated with the decreasing flux of gas and ash throughout the vent that is consistent with the waning of the pyroclastic column.

After the first 3-4 days of the eruption, changes in eruptive activity were evident (Pistolesi et al., 2015; Castro et al., 2013, 2016; Schipper et al., 2021) with a rapidly decreasing height of the pyroclastic column and the dominance of ballistic ejection of bombs. This was associated with the intrusion of a laccolith at very shallow depths (<200 m) underneath the eruptive vent (Castro et al., 2016). Afterwards, the eruption has been described as having a hybrid explosive-effusive character (Castro et al., 2013, 2016; Schipper et al., 2021; Delgado, 2021), although we preferred to separate an initial explosive phase from a dominantly effusive phase that started roughly 10 days after the eruption onset. The beginning of the effusive phase is clearly marked by the first occurrence of a quasi-harmonic tremor that starts dominating the seismic signal since June 15th. This anticipated the visual observation of a lava flow some days after. The modulated character of the signal, its large amplitude (several times larger than the spasmodic tremor of the explosive phase) and shallow source region were interpreted by Bertin et al. (2015) as indicative of oscillations of volcanic layers by the pass of moving magma toward the surface. However alternative processes have been proposed to explain (quasi-)harmonic tremors accompanying other eruptions, as the cyclic replenishment of the shallow magma chamber at Etna (La Delfa et al., 2001) or the oscillation of gas bubbles escaping from the erupting magma at Etna (Alperone et al., 2003) and Vulcano (Montegrosi et al., 2019). In the case of Cordon Caulle this latter possibility can be partially supported by the model of Shipper et al. (2021) for which magma outgassing and ash sintering in the conduit are in a constant competition causing the fluctuations in the eruptive-effusive character of the 2011 eruption.

In this context, another relevant question stated by the NASEM17 report is: What are the critical thresholds in processes and physical properties that govern shifts in eruptive behavior? Castro et al. (2016) and Schipper et al. (2021) have proposed that the shift from purely explosive to hybrid explosive-effusive behavior of the 2011 PCCVC eruption didn´t involve a reduction of magma fragmentation efficiency and shift from close to open system degassing as classically supposed (i.e. Eichelberger et al., 1986; Giachetti et al., 2020), but it is mainly controlled by the simultaneous occurrence of self-extinguishing explosivity by viscous sintering of fine pyroclastic material inside the conduit and the arrival of coherent melt at the vent. Such a complex

behavior is thought to be influenced by the strong structural control of PCCVC, which allows tapping different parts of the large silicic reservoir following different pathways to the vent and suffering different physical processes (Castro et al., 2013, 2016; Alloway et al., 2015). Our results partially complement this model by showing compressive focal mechanisms near the vent during the change from explosive to hybrid behavior, which is consistent with local constriction/blockage of conduit and the initiation of laccolith intrusion. Interestingly, this near-vent compression was coeval with the regional relaxation of the system as shown by InSAR-determined deflation covering the outside limits of PCCVC and peaking *b-values* at the transition between explosive and effusive phases (Fig. 3E). Very high rates of magma discharge at the beginning of the effusive phase are then followed by a gradual decrease, which is mimicked by changes in the amplitude of the quasi-harmonic tremor. This is marked also by renewed VT-HB seismicity with relatively larger magnitudes implying a further decrease of *b-value* and therefore the transient stressing of the system around the conduit as blockage and sintering continue to be dominant. This is simultaneous with regional subsidence but associated with smaller deflation volumes determined by InSAR and a larger area and depth extent of the volume of rock affected by seismicity.

The second half of the effusive phase and the declining phase show transit to waning stages of the eruption, as measured by decreasing effusion rates (Bertin et al., 2015). In this regard, the NASEM17 report establishes the question as to why do volcanoes stop erupting, and how do we recognize when an eruptive episode is over? The most likely process accompanying this waning stage is the reduction of the magma supply rate feeding the eruption, which is consistent with a decrease in the amplitude of the quasi-harmonic tremor, smaller deflation volumes of the deep source recognized by InSAR (Jay et al., 2014; Wendt et al., 2016, Delgado et al., 2019), and a return of *b-value* to levels observed before the eruption. The latter is associated to a smaller amount of VT-HB seismic events compared with previous phases and a larger magnitude range, coeval with an expansion of seismicity to the outer limits of PCCVC. This includes a further extension of seismicity along the LOFZ main branch southward of Puyehue volcano with focal mechanism showing a dominant strike-slip motion (Fig. 8). Although this behavior could have been interpreted as anticipating the end of the eruption, the resurging phase showed renewed seismicity mostly of LP origin (that we interpret as a synthon of a new pulse of magma rising through the conduit) with swarms of VT-HB seismicity located either near main magma sources tapped during the previous phases or at regional faults. The occurrence of these swarms is associated with a gradual increase of *b-value* pointing to a further relaxation of the system and coincides with a renewed energy of quasi-harmonic tremor that further correlates with slightly augmented rates of magma discharge. Our study finished in December 2011, but the effusion of lava and very minor pyroclastic events continued with progressively waning activity until March 2012 (Castro et al., 2013; Delgado, 2021; Schipper et al., 2021).

**Acknowledgements**

We thank the Observatorio Volcanológico de Los Andes del Sur OVDAS and SERNAGEOMIN for facilitating the data to carry out this research. This work was partially funded by the Chilean National Agency for Research and Development

(ANID), Postdoctoral Program 2020-2022/N°3200387, FONDEF Project: N°19I10397; Fondecyt Project 1151175 Active Tectonics and Volcanism at the Southern Andes (ACT&VO), Millennium Nucleus NCN19_167 The Seismic Cycle along

Subduction Zones (CYCLO), and Project: CIVUR-39: "Centro Interactivo Vulcanológico de La Araucanía" N°FRO2193, Desarrollo de Actividades de Interés Nacional (ADAIN), Ministerio de Educación, Gobierno de Chile. We would also like to thank the sponsoring institution Universidad de La Frontera (Department of Civil Engineering) and GeoAraucanía Group, a university community made up of professionals linked to geosciences, forming part of the faculties of Engineering of the Universidad de la Frontera (UFRO), Engineering of the Catholic University of Temuco (UCT) and Science, Engineering and

Technology of the Universidad Mayor, whose objective is to develop and promote sustainable lines of research linked to the territory and its communities.

## Data Availability

The original seismic record used in this study belongs to the Observatorio Volcanológico de los Andes del Sur (OVDAS) that is part of the Chilean Geological Survey (Servicio Nacional de Geología y Minería, SERNAGEOMIN). Access to the original

data must be requested from this institution (https://www.portaltransparencia.cl/PortalPdT/ingreso-sai-v2?idOrg=undefined).

## Author Contribution

DB and AT conducted the research. JL, LF, CC and JSM organized the database and contributed to the primary analysis of seismic activity. AT, FG, CF and MC contributed to the discussion of the results, suggesting different strategies to carry out a complete analysis of the seismic activity. AT and MC design and performed the figures. AT and DB wrote the paper. All

authors contributed to manuscript revision, read, and approved the submitted version.

## Competing Interests

The authors declare no competing interests.

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
