# Peer review of "Anatomy of a High-Silica Eruption as Observed by a Local Seismic Network: The June 2011 Puyehue-Cordón Caulle Event (Southern Andes, Chile)"

_EGUsphere, 2022_

## Referee Comment (RC1)

This paper presents an analysis of the seismic activity of the Cordón Caulle volcano before and during its large eruption of 4 June, 2011. The data processing includes mainly the detection and classification of the seismic events, the hypocentre relocation using a refined velocity model, the determination of focal mechanism of the main events and the evaluation of the b-value as a function of time. Using the results of this data processing and other types of observations, a detailed chronology of the unrest and eruptive sequences, divided in 7 phases, is proposed.

The seismic activity during the unrest and pre-eruptive periods is interpreted in light of the 'top-down model' of precursory seismicity by Roman and Cashman (2018). During the eruption itself, several phases, explosive, effusive, declining and resurging, are identified and the corresponding behaviour of the seismicity is analysed.

It is important to present the seismic observations associated with such a large eruptive event. The manuscript is generally well written and clear. It will deserve publication in SE when two main issues and some minor ones are clarified.

First, a large part of the analysis and interpretation is based on the temporal variations of the b-value. According to fig. 2D, this parameter varies between 0.3 and 1.2 approximately, which is a huge interval that I've never seen in the literature. The largest variations occurred in a few days at the beginning of the eruption. In contrast with the unrest period, the eruptive phase is characterized by a strong tremor which may have modified the magnitude completeness Mc of the catalogue. This point should be discussed thoroughly. I suggest the authors to use more robust methods that are not dependent on the estimation of Mc. For example:

B-Positive: A Robust Estimator of Aftershock Magnitude Distribution in Transiently Incomplete Catalogs
van der Elst, NJ
JOURNAL OF GEOPHYSICAL RESEARCH-SOLID EARTH
Volume126, issue2, Article Numbere2020JB021027
DOI10.1029/2020JB021027
Published FEB 2021

Inverse Migration of Seismicity Quiescence During the 2019 Ridgecrest Sequence
Marsan, D ; Ross, ZE
JOURNAL OF GEOPHYSICAL RESEARCH-SOLID EARTH
Volume126, Issue3, Article Numbere2020JB020329
DOI10.1029/2020JB020329
Published MAR 2021

Furthermore, the authors interpret the b-value variations as due to stress variations in the structure. However, if we use the relationship found by Scholz (2015) between b and s1-s3, we'd calculate variations of several hundreds of MPa for the deviatoric stress associated to the b-value variations obtained in the present paper. This point should be clarified and its physical implication should be discussed.

Second, the authors discuss the behaviour of the harmonic tremor that appeared during the effusive phase. They refer to a previous paper of their research group (Bertin et al., 2015) which presents a very elementary analysis of particle motion of the tremor and which indicates that horizontally

polarized P-waves are dominant in the wavefield. This observation is very intriguing. They conclude that the source is very shallow and due to the oscillation of volcanic layers excited by the magma flow. This conclusion and interpretation are poorly founded. The wave polarization analysis should be carried out with much more details and using several stations, following for example the approach of Haney et al. (2020). Alternative source mechanisms should be discussed because the model of Omer (1950) can hardly account for harmonic tremor with regularly spaced spectral peaks.

Third, the final unrest phase and the onset of the eruption are critical periods. I suggest to present an enlargement of fig. 2 from the end of May to beginning of June in order to present with more details the behaviour of the seismic activity and other parameters of interest.

Minor comments.

Figure 1. The labels of the main map and of the inset are mixed. Separate them and indicate that red rectangle shows limits of main map.

Line 132-134. The results of the analysis of wave polarization with Matsumura's method are not presented in the paper. Show them.

Line 156. The sentence is not clear.

Line 169. The magnitude of completeness may have changed during the study period. In order to convince the readers that the b-value is well defined, several Gutenberg-Richter diagrams should be displayed for large, medium or small b-values. See also first main comment.

Line 334. Are the depths of 3.8-6.4 km the depths of the 2 point sources or a range of depth common of these sources?

Fig 5 and other. Squares and crosses representing polarities in beach balls are not visible.

394. Suppress 'during'.

403. Explain your arguments.

452. Suppress 'Oscillation of'.

489. Replace 'seismicity' by 'events'.

538. The top-down model is verified for volcanoes after repose intervals of some decades. This is the case of Cordon Caulle. This could be mentioned.

599 & 636. Into → In

628. Replace 'has' by 'as'.

660. 'a stabilization of b-value to pre-eruptive conditions': what do you mean?

665-667. How do you interpret this LP seismicity in September-November?

785. pages are 221-239.

826. Singer or Asinger?

835. 2016

---

## Referee Comment (RC2)

**Overview**

This manuscript presents a detailed chronology of the June 2011 Cordon Caulle eruption, through careful analysis of near- and far-field seismicity. This particular eruption has been extensively studied in terms of the petrology (Jay et al., 2014) and deformation (Wendt. et al., 2016, Delgado et al., 2019), however, this is the first presentation of the seismic sequence. The authors make an excellent case for the motivations and integrity of the work, specifically in Section 5.2 referencing the importance of geophysical monitoring of unrest in the NASME17 Report.

The eruption is defined by a 7-stage process, defined by marked changes in the seismicity which then correspond to previous models from InSAR and GNSS observations. The focus for this chronology is March to December 2011, although the authors do acknowledge unrest outwith this time frame. The authors make use of over 30,000 manually identified earthquakes and go on to include analysis of event rates, b-values, amplitudes, reduced displacement and where possible hypocentral locations and focal mechanisms.

The discussion section is structured heavily on the 'top down' model presented by Roman & Cashman. This creates a flowing and well-structured narrative for the eruption sequence and ties the manuscript nicely together to conclude. Overall, I would recommend this manuscript for publication subject to a few revisions and points to consider.

**Major points**

My main issue that I found was in the early part of the manuscript and is to do with volcanic seismicity and terminology. This whole study hinges on the understanding and interpretation of the seismicity, and particularly where certain 'types' of seismicity dominate the signal. Therefore, I think the manuscript would benefit from a careful and explicit description of how these categories are defined. In line 124 the authors suggest that earthquakes can be manually labelled based on their "waveform appearance", and then in section 3.2 the authors introduce volcano seismic event labels and classifications, such as VTs, LPs, HBs and VLPs which are used repeatedly through the study. I would suggest that this section is expanded to include a more careful description of these event types. For example, I think there is a much better definition of a HB earthquake on line 271. For LPs, it would help to describe their waveform appearance, including details like their emergent onset and long single frequency coda tails. There needs to be some care taken when using the terms low-frequency (LF) vs. long-period (LP) and their association with fluid movement. For instance, some studies have shown LPs are not strictly always associated with fluid movement (*Bean, C., De Barros, L., Lokmer, I. et al. Long-period seismicity in the shallow volcanic edifice formed from slow-rupture earthquakes. Nature Geosci 7, 71–75 (2014). https://doi.org/10.1038/ngeo2027*).

My second main point related to this, is to do with tremor. Particularly in the latter stages of the chronology and the discussion section, there is a lot of interpretation associated with spasmodic, emission and harmonic/quasi-harmonic tremor. I think if these are going to be referenced then they should be outlined and described in this early section 3.2 with the other seismicity event types. Line 129 suggests that tremor has to have a frequency content <5 Hz but this is not always true. Later in sections 4.3.2 and 4.3.3 the authors describe the

frequency content and amplitudes of the tremor. From figure 2, the tremor appears to be occurring during a period where there are still hundreds of individual earthquakes per day. I think the methods section would benefit from a sentence or two describing how the authors isolated the tremor from discrete events and analysed it. For example, how is the tremor classified and labelled to be included in Figure 2? Because some quite major conclusions are derived about effusion rates correlating with tremor amplitudes, so it is important to be clear about how exactly this is done.

My last main point, would be to recommend the inclusion of more figures in the main text (provided that this is not a limitation from the journal or in print). There are more figures in the supporting material than the main text. A simple diagram illustrating a "typical" LP, VT, HB and tremor would really support the added detail that I have previously recommended. The reader can then see what these earthquakes really look like. Figures S1, 3, 4 and 5 could all be adapted to make a conglomerate main text figure. The authors also describe a very detailed final model for where they believe the magma storage and seismicity is originating in this eruption. I think that a cartoon diagram would support this conceptual model. It does not need to be a mapped tomographic map, but something to visually describe the volcanic system, the magma storage, the inflation and deflation lobes, the locations of seismicity during different phases etc. would all really help bring the conclusions together.

**Minor points and line comments**

- Line 40: "which means, however, that almost…" insert commas
- In paragraph two, the authors refer to triggered volcanic unrest following the 1960 earthquake but then never really circle back to this idea again. It would be good to get reference and a comment on this in the conclusions or discussion.
- Line 66: "The volcanic complex is composed of three main…"
- Figure 1: Despite looking through the text, I can't see what the acronym CLVVC stands for
- Line 106: Change unprecedented to something like 'novel' or 'significant'.
- Figure 2: Panel D error bars, but what kind of error? Standard deviation? Standard error? Some other measure?
- Line 168: How did you decide the magnitude of completeness? Did you calculate it or decide on a cut off threshold yourself? Do you have a plot to justify this? It could be a supplementary figure.
- Line 224: Change to something like 'This trend continued in April when episodes of LP and HB seismicity lasted several days…'. The word pulses has a dynamic connotation that suggests it's coming from a source.
- Line 227: April
- Line 230: I would choose a different word from crowned. It's used a few times in the text. Maybe "culminated", or "reached a maximum"
- Figures 3-6 all need graphical keys as well as description in the text. I still can't really work out what the colours of the stars and squares in the location plots relate to.
- Figure 3 caption: typo – stars/starts
- When reading the Final Unrest Phase and Eruption Onset, figure 2 becomes too hard to read details. It would be good to include a similar figure but zoomed in to only this week long period so the reader can see the details for themselves.
- Line 294: Rephrase and do not use the word coronating (same as crowning, above).

- Line 449: Are you calculating the magnitudes of LP events. If so, how? Without clear onsets or S- phases.
- Line 485: "During the second half of September, an increase of LP seismicity was recorded." – can you elaborate on this sentence at all? Or include some quantities?
- Line 567: This is the first mention of VLPs since the section 3.2. I appreciate this is before the timeline of the study but can you either remove this comment or add another sentence to explain why it is significant to have observed VLPs at that time?
- Line 576: Another
- Line 592: This section begins a discussion about the mobilization of fluids, but with no reference to LP seismicity. In section 3.2 there is specific reference to LPs being associated with fluid movement, so I think that should be mentioned here somewhere too.
- Take care when referencing White & McCausland in section 5, particularly in parallel to Roman & Cashman. White & McCausland actually describes more of a bottom-up process in many ways, and does not completely agree with Roman & Cashman. Be explicit when you reference these as to which bits of your chronology relate to which sections of their conceptual models.

---

## Author Comment (AC1)

RESPONSE TO RC1
Original comments of RC1 are in black below and our answers are highlighted in blue

This paper presents an analysis of the seismic activity of the Cordón Caulle volcano before and during its large eruption of 4 June, 2011. The data processing includes mainly the detection and classification of the seismic events, the hypocentre relocation using a refined velocity model, the determination of focal mechanism of the main events and the evaluation of the b-value as a function of time. Using the results of this data processing and other types of observations, a detailed chronology of the unrest and eruptive sequences, divided in 7 phases, is proposed.

The seismic activity during the unrest and pre-eruptive periods is interpreted in light of the 'top-down model' of precursory seismicity by Roman and Cashman (2018). During the eruption itself, several phases, explosive, effusive, declining and resurging, are identified and the corresponding behaviour of the seismicity is analysed.

It is important to present the seismic observations associated with such a large eruptive event. The manuscript is generally well written and clear. It will deserve publication in SE when two main issues and some minor ones are clarified.

We are grateful to RC1 for his dedicated review and positive evaluation of our manuscript. Below we assess each of his comments and suggestions, most of which were implemented in the new version and greatly improved the clarity and quality of the paper.

First, a large part of the analysis and interpretation is based on the temporal variations of the b-value. According to fig. 2D, this parameter varies between 0.3 and 1.2 approximately, which is a huge interval that I've never seen in the literature. The largest variations occurred in a few days at the beginning of the eruption. In contrast with the unrest period, the eruptive phase is characterized by a strong tremor which may have modified the magnitude completeness Mc of the catalogue. This point should be discussed thoroughly. I suggest the authors to use more robust methods that are not dependent on the estimation of Mc. For example:

B-Positive: A Robust Estimator of Aftershock Magnitude Distribution in Transiently Incomplete Catalogs
van der Elst, NJ
JOURNAL OF GEOPHYSICAL RESEARCH-SOLID EARTH
Volume126, issue2, Article Numbere2020JB021027
DOI10.1029/2020JB021027
Published FEB 2021

Inverse Migration of Seismicity Quiescence During the 2019 Ridgecrest Sequence
Marsan, D ; Ross, ZE
JOURNAL OF GEOPHYSICAL RESEARCH-SOLID EARTH
Volume126, Issue3, Article Numbere2020JB020329
DOI10.1029/2020JB020329
Published MAR 2021

We greatly appreciate this comment of R1 that stimulate us to reanalyze the computation and interpretation of the b-value time series. As suggested, we implemented the b-positive method of van der Elst (2021) considering the positive magnitude differences of successive seismic events instead of their actual magnitudes. The new b-value time series replaces the old one in panel E of the new figure 3 (old Fig. 2). As seen in the figure below, both time series are generally similar in their temporal evolution, mostly after the explosive/effusive transition.

[Figure]

The most significant differences can be appreciated between the final unrest phase and after the eruption onset: in particular, the rapid decrease of b-value right before the eruption to unrealistic values near 0.2 is no longer visible and we think that this was an artifact caused by the increase of the magnitude of completeness Mc as the number of large earthquakes was rapidly increased. We still observe a rapid decrease of b before the eruption and then a recover to higher values but this is preceded by an increase of b at the end of the enhanced unrest phase.

We modified the description of the method for computing the b-value time series in section 3.3.3. Given the differences between both b-value time series, mostly before the eruption, we also modified the description of results along section 4.

Furthermore, the authors interpret the b-value variations as due to stress variations in the structure. However, if we use the relationship found by Scholz (2015) between b and s1-s3, we'd calculate variations of several hundreds of MPa for the deviatoric stress associated to the b-value variations obtained in the present paper. This point should be clarified and its physical implication should be discussed.

Based on the new b-value time series we now include an explicit discussion in section 5.2 and 5.3 regarding changes of differential stress using the relationship of Scholz (2015). Since temporal b-value variations are relatively smooth in our new version, estimated changes in differential stress are quite realistic between <50 to 300-400 MPa and help to understand the general shift of the system from compression before the eruption to relaxation after its onset.

Second, the authors discuss the behaviour of the harmonic tremor that appeared during the effusive phase. They refer to a previous paper of their research group (Bertin et al., 2015) which presents a very elementary analysis of particle motion of the tremor and which indicates that horizontally polarized P-waves are dominant in the wavefield. This observation is very intriguing. They conclude that the source is very shallow and due to the oscillation of volcanic layers excited by the magma flow. This conclusion and interpretation are poorly founded. The wave polarization analysis should be carried out with much more details and using several stations, following for example the approach of Haney et al. (2020). Alternative source mechanisms should be discussed

because the model of Omer (1950) can hardly account for harmonic tremor with regularly spaced spectral peaks.

We are also grateful to this comment that allows us to perform a deeper analysis of the quasi-harmonic tremor signal. We follow the method of Matsumura (1981) and performed a polarization analysis of stations PHU, QIR and ANT that complement and expand the analysis of Bertin et al. (2015). This analysis reinforces their main conclusion as to that the polarization azimuth is sub-parallel to the direction between each station and the vent. We include a new figure in the Supplementary Material showing the results of this analysis and modified the text in section 4.3.3 in accordance. We also discuss alternative source mechanisms for the tremor than Omer (1950) in section 5.3.

Third, the final unrest phase and the onset of the eruption are critical periods. I suggest to present an enlargement of fig. 2 from the end of May to beginning of June in order to present with more details the behaviour of the seismic activity and other parameters of interest.

We agree with R1 that an enlargement of the critical period right before and after the eruption onset can be beneficial for the reader. Therefore, we created a new figure 3 showing the time series for number of events per day, local magnitude, b-value and location depth for the time window may 21 to June 15.

Minor comments.
Figure 1. The labels of the main map and of the inset are mixed. Separate them and indicate that red rectangle shows limits of main map.
Done

Line 132-134. The results of the analysis of wave polarization with Matsumura's method are not presented in the paper. Show them.
We now include these results as part of the supplementary information.

Line 156. The sentence is not clear.
Done

Line 169. The magnitude of completeness may have changed during the study period. In order to convince the readers that the b-value is well defined, several Gutenberg-Richter diagrams should be displayed for large, medium or small b-values. See also first main comment.
As explained before, we now implemented the b-positive method of van der Elst (2021), which eliminates the problem of changing Mc through time, therefore we think that including GR diagrams is not necessary.

Line 334. Are the depths of 3.8-6.4 km the depths of the 2 point sources or a range of depth common of these sources?
This was confusing, so we modified the text: The source of deformation has been modeled by two deflating point sources at depths of 3.8 and 6.1 km (Jay et al., 2014, Wendt et al., 2016).

Fig 5 and other. Squares and crosses representing polarities in beach balls are not visible.

We do not agree and think that these symbols are sufficiently visible. Modifying these figures will be too much work for only a minor improvement of the visibility of these symbols, something that we would like to avoid

394. Suppress 'during'.
Done

403. Explain your arguments.
Following your suggestion as commented before, we now include a deeper analysis of the tremor polarization in this section.

452. Suppress 'Oscillation of'.
Done

489. Replace 'seismicity' by 'events'.
Done

538. The top-down model is verified for volcanoes after repose intervals of some decades. This is the case of Cordon Caulle. This could be mentioned.
Done

599 & 636. Into by In
Done

628. Replace 'has' by 'as'.
Done

660. 'a stabilization of b-value to pre-eruptive conditions': what do you mean?
That the b returned to values similar to those registred before the eruption. We clarified this in the text.

665-667. How do you interpret this LP seismicity in September-November?
As a new pulse of magma rising through the conduit. We include this in the text

785. pages are 221-239.
Done

826. Singer or Asinger?
Singer of course, thanks!

835. 2016
Done

---

## Author Comment (AC2)

REVIEWER 2
Overview
This manuscript presents a detailed chronology of the June 2011 Cordon Caulle eruption, through careful analysis of near- and far-field seismicity. This particular eruption has been extensively studied in terms of the petrology (Jay et al., 2014) and deformation (Wendt. et al., 2016, Delgado et al., 2019), however, this is the first presentation of the seismic sequence. The authors make an excellent case for the motivations and integrity of the work, specifically in Section 5.2 referencing the importance of geophysical monitoring of unrest in the NASME17 Report.
The eruption is defined by a 7-stage process, defined by marked changes in the seismicity which then correspond to previous models from InSAR and GNSS observations. The focus for this chronology is March to December 2011, although the authors do acknowledge unrest outwith this time frame. The authors make use of over 30,000 manually identified earthquakes and go on to include analysis of event rates, b-values, amplitudes, reduced displacement and where possible hypocentral locations and focal mechanisms.
The discussion section is structured heavily on the 'top down' model presented by Roman & Cashman. This creates a flowing and well-structured narrative for the eruption sequence and ties the manuscript nicely together to conclude. Overall, I would recommend this manuscript for publication subject to a few revisions and points to consider.

We appreciate the thoughtful review and positive comments of our manuscript done by RC2. Below we assess each of his comments and suggestions, most of which were implemented in the new version and greatly improved the clarity and quality of the paper.

Major points
My main issue that I found was in the early part of the manuscript and is to do with volcanic seismicity and terminology. This whole study hinges on the understanding and interpretation of the seismicity, and particularly where certain 'types' of seismicity dominate the signal. Therefore, I think the manuscript would benefit from a careful and explicit description of how these categories are defined.

We agree with R2 that a better description of each type of volcanic seismic signal is necessary at the beginning of section 3.2 and therefore we expanded this section in the new version as described below. This includes adding a new figure as also suggested by R2 showing examples of the seismic record for each type of signal.

In line 124 the authors suggest that earthquakes can be manually labelled based on their "waveform appearance", and then in section 3.2 the authors introduce volcano seismic event labels and classifications, such as VTs, LPs, HBs and VLPs which are used repeatedly through the study. I would suggest that this section is expanded to include a more careful description of these event types. For example, I think there is a much better definition of a HB earthquake on line 271.

We follow this suggestion and move text of the original line 271 to the description of HB events in section 3.2.

For LPs, it would help to describe their waveform appearance, including details like their emergent onset and long single frequency coda tails.

We included more information regarding the appearance of the waveform for LP events.

There needs to be some care taken when using the terms low-frequency (LF) vs. long-period (LP) and their association with fluid movement. For instance, some studies have shown LPs are not strictly always associated with fluid movement (Bean, C., De Barros, L., Lokmer, I. et al. Long-period seismicity in the shallow volcanic edifice formed from slow-rupture earthquakes. Nature Geosci 7, 71–75 (2014). https://doi.org/10.1038/ngeo2027).

We included a mention to a possible alternative interpretation of LP events citing the work of Bean et al. (2014).

My second main point related to this, is to do with tremor. Particularly in the latter stages of the chronology and the discussion section, there is a lot of interpretation associated with spasmodic, emission and harmonic/quasi-harmonic tremor. I think if these are going to be referenced then they should be outlined and described in this early section 3.2 with the other seismicity event types. Line 129 suggests that tremor has to have a frequency content <5 Hz but this is not always true. Later in sections 4.3.2 and 4.3.3 the authors describe the frequency content and amplitudes of the tremor.

We agree with R2 and now included a detailed description of the two identified tremors in section 3.2 along with an example of the seismic record of both tremor types in the new Fig. 2.

From figure 2, the tremor appears to be occurring during a period where there are still hundreds of individual earthquakes per day. I think the methods section would benefit from a sentence or two describing how the authors isolated the tremor from discrete events and analysed it. For example, how is the tremor classified and labelled to be included in Figure 2? Because some quite major conclusions are derived about effusion rates correlating with tremor amplitudes, so it is important to be clear about how exactly this is done.

We include now in section 3.2 an explanation on how the low-frequency tremor signal was isolated from the high-frequency VT and HB events by using an appropriated Butterword filter.

My last main point, would be to recommend the inclusion of more figures in the main text (provided that this is not a limitation from the journal or in print). There are more figures in the supporting material than the main text. A simple diagram illustrating a "typical" LP, VT, HB and tremor would really support the added detail that I have previously

recommended. The reader can then see what these earthquakes really look like. Figures S1, 3, 4 and 5 could all be adapted to make a conglomerate main text figure.

We agree with R2 regarding the possible inclusion of more figures in the text to support for instance the description of the different seismic signals, and therefore we now included a new Fig. 2 with the examples that were originally in the supplementary material.

The authors also describe a very detailed final model for where they believe the magma storage and seismicity is originating in this eruption. I think that a cartoon diagram would support this conceptual model. It does not need to be a mapped tomographic map, but something to visually describe the volcanic system, the magma storage, the inflation and deflation lobes, the locations of seismicity during different phases etc. would all really help bring the conclusions together.

We agree that a conceptual scheme integrating the multiple aspects of the spatio-temporal evolution of the magmatic plumbing systems can be very useful in order to clarify what is described in the discussion. We made an effort to create such a figure trying to integrate our results with those of previous authors to describe the crustal structure of the magmatic plumbing system (new Fig. XX) and the main phases of its evolution (Fig. XXX).

Minor points and line comments
• Line 40: "which means, however, that almost…" insert commas
Done

• In paragraph two, the authors refer to triggered volcanic unrest following the 1960 earthquake but then never really circle back to this idea again. It would be good to get reference and a comment on this in the conclusions or discussion.
As the main results and the discussion of the paper really deals with the evolution of this particular eruption, we think it is complicated to come back to a potential triggering of the 2011 eruption by the 2010 Maule earthquake, something that cannot be supported or discarded by our data. We prefer to not mention this in the discussion

• Line 66: "The volcanic complex is composed of three main…"
?

• Figure 1: Despite looking through the text, I can't see what the acronym CLVVC stands for
It stands for Carran-Los Venados Volcanic Complex, which now we added to the caption of Fig. 1

• Line 106: Change unprecedented to something like 'novel' or 'significant'.
Done

• Figure 2: Panel D error bars, but what kind of error? Standard deviation? Standard error? Some other measure?
This are errors computed using the approach of Aki (1965) as described in section 3.3.3.

• Line 168: How did you decide the magnitude of completeness? Did you calculate it or decide on a cut off threshold yourself? Do you have a plot to justify this? It could be a supplementary figure.
Following a suggestion of R1, we implemented the b-positive method of van der Elst (2021) for computing the b-value, which eliminates the potential problem of a time-variable Mc as now explained in section 3.3.3.

• Line 224: Change to something like 'This trend continued in April when episodes of LP and HB seismicity lasted several days…'. The word pulses has a dynamic connotation that suggests it's coming from a source.
Done

• Line 227: April
Done

• Line 230: I would choose a different word from crowned. It's used a few times in the text. Maybe "culminated", or "reached a maximum"
Done

• Figures 3-6 all need graphical keys as well as description in the text. I still can't really work out what the colours of the stars and squares in the location plots relate to.
We expanded the explanation of colors of starts and squares in the caption of new Fig. 4 (old Fig. 3).

• Figure 3 caption: typo – stars/starts
Done

• When reading the Final Unrest Phase and Eruption Onset, figure 2 becomes too hard to read details. It would be good to include a similar figure but zoomed in to only this week long period so the reader can see the details for themselves.
This was also a suggestion of R1. We now include a new Fig. 4 with an enlaregement of the time window between May 21 and June 15.

• Line 294: Rephrase and do not use the word coronating (same as crowning, above).
Done

• Line 449: Are you calculating the magnitudes of LP events. If so, how? Without clear onsets or S- phases.
The sentence was misleading since we refered to the magnitude of the VT events. We modified it in the text

• Line 485: "During the second half of September, an increase of LP seismicity was recorded." – can you elaborate on this sentence at all? Or include some quantities?
Done

• Line 567: This is the first mention of VLPs since the section 3.2. I appreciate this is before the timeline of the study but can you either remove this comment or add another sentence to explain why it is significant to have observed VLPs at that time?
We replaced VLP by LP

• Line 576: Another
Done

• Line 592: This section begins a discussion about the mobilization of fluids, but with no reference to LP seismicity. In section 3.2 there is specific reference to LPs being associated with fluid movement, so I think that should be mentioned here somewhere too.
LP events were not recognized during the Enhanced Unrest phase, but we explicit in this sentence now that "opening of fractures" is related to VTs and "mobilization of magma and fluids" is related to HBs.

• Take care when referencing White & McCausland in section 5, particularly in parallel to Roman & Cashman. White & McCausland actually describes more of a bottom-up process in many ways, and does not completely agree with Roman & Cashman. Be explicit when you reference these as to which bits of your chronology relate to which sections of their conceptual models.
We think that the mention of White and McCausland are correct since we refer to specific features of volcano-tectonic seismic activity described by them and not to a conceptual model supported by these authors.